# Integrated dual-laser photonic chip for high-purity carrier generation enabling ultrafast terahertz wireless communications

Shi Jia [1,9✉], Mu-Chieh Lo [2,3,9], Lu Zhang [4,5,9], Oskars Ozolins [4,6,7], Aleksejs Udalcovs [6], Deming Kong [1], Xiaodan Pang [4✉], Robinson Guzman [2], Xianbin Yu [5], Shilin Xiao [8], Sergei Popov [4], Jiajia Chen [4], Guillermo Carpintero [2✉], Toshio Morioka [1], Hao Hu [1✉] & Leif K. Oxenløwe [1]

Photonic generation of Terahertz (THz) carriers displays high potential for THz communications with a large tunable range and high modulation bandwidth. While many photonics-based THz generations have recently been demonstrated with discrete bulky components, their practical applications are significantly hindered by the large footprint and high energy consumption. Herein, we present an injection-locked heterodyne source based on generic foundry-fabricated photonic integrated circuits (PIC) attached to a uni-traveling carrier photodiode generating high-purity THz carriers. The generated THz carrier is tunable within the range of 0–1.4 THz, determined by the wavelength spacing between the two monolithically integrated distributed feedback (DFB) lasers. This scheme generates and transmits a 131 Gbits$^{-1}$ net rate signal over a 10.7-m distance with −24 dBm emitted power at 0.4 THz. This monolithic dual-DFB PIC-based THz generation approach is a significant step towards fully integrated, cost-effective, and energy-efficient THz transmitters.

---

[1] DTU Fotonik, Technical University of Denmark, DK-2800, Kgs, Lyngby, Denmark. [2] Universidad Carlos III de Madrid, 28911 Leganés, Madrid, Spain. [3] Optical Networks Group, University College London, London WC1E 7JE, UK. [4] KTH Royal Institute of Technology, 164 40 Kista, Sweden. [5] College of Information Science and Electrical Engineering, Zhejiang University, 310027 Hangzhou, China. [6] RISE Research Institutes of Sweden, 164 40 Kista, Sweden. [7] Institute of Telecommunications, Riga Technical University, Riga LV-1048, Latvia. [8] School of SE-IEE, Shanghai Jiao Tong University, 200240 Shanghai, China. [9] These authors contributed equally: Shi Jia, Mu-Chieh Lo, Lu Zhang. ✉email: shijai@fotonik.dtu.dk; xiaodan@kth.se; guiller@ing.uc3m.es; huhao@fotonik.dtu.dk

Today's global wireless data traffic grows by ~50% per year, and existing wireless communication in traditional lower radio frequency (RF) bands faces significant challenges to meet the demand of this explosive growth[1–3]. Higher RF bands providing broader bandwidth need to be explored to support ultrafast wireless communications. The terahertz range (THz, 0.3–10 THz) is promising to fill the data rate gap between fiber optic and wireless networks. The generation of THz carriers using photonic techniques is interesting because of the large tuning range and high modulation bandwidth, enabling the generation of high-quality THz signals capable of carrying 100 Gbits$^{-1}$ data rates and beyond[4].

Photonic integrated circuit (PIC) based THz synthesizers have the advantages of low weight, small footprint, and low power consumption. Moreover, facilitated by the continuous development of semiconductor fabrication technologies, the open-access InP generic foundry photonic integration approach has allowed active and passive components to be monolithically integrated on the same substrate[5]. Therefore, combining two single-wavelength distributed feedback (DFB) lasers in parallel with a coupler, the open foundry platform has recently proven the potential for heterodyne THz generation at frequencies >1.3 THz, potentially reducing the cost significantly. However, the tens-of-MHz laser linewidth and the lack of phase correlation between two free-running lasers lead to considerable instability of the heterodyne THz signals, limiting practical applications[6].

In terms of THz wireless communication reach, the 10–100 m range is recommended for indoor communications for carrier frequencies ranging from 350 to 910 GHz[2]. Extensive research has been conducted to achieve high data rate transmissions at THz frequencies alongside the efforts to reach longer distances. Figure 1 summarizes the experimentally demonstrated data rates and wireless reaches in the 350–910 GHz band[2,7–15]. With photonics-based schemes, high data rates can be achieved by frequency division multiplexing techniques, often yielding data rates of 100 Gbits$^{-1}$ and above[8–12]. However, these multiplexing techniques increase the system cost and complexity. Single-channel wireless transmission with a data rate beyond 100 Gbits$^{-1}$ has recently been demonstrated over a very short distance[15].

In this paper, we demonstrate a single-channel THz photonic-wireless transmission achieving both high data rate and relevant reach, with >100 Gbits$^{-1}$ data rate and >10 m reach using a PIC-based THz transmitter. Monolithically integrated dual DFB lasers are injection-locked by a mode-locked laser (MLL) based optical frequency comb[16]; therefore, the generated THz carrier is phase coherent and stable. A wavelength division demultiplexer separates the generated two continuous wave (CW) tones. One of the CW tones is modulated with a high-speed signal and then beats with the other unmodulated CW tone in a photo-mixing uni-traveling carrier photodiode (UTC-PD). Finally, the modulated signal with a net rate of 131 Gbits$^{-1}$ on the 0.4-THz carrier is emitted through a THz antenna. After 10.7 m wireless transmission, the THz signal is received by a THz receiver. To the best of our knowledge, this is the highest data rate for a single-channel THz wireless transmission and the largest bandwidth-distance product for the carrier frequencies above 350 GHz. In addition, compared with previous works, a significantly lower emitted THz power/bitrate/distance of $2.9 \times 10^{-18}$ J·bit$^{-1}$·m$^{-1}$ has been achieved, indicating the superior noise performance of the generated THz carrier. Table 1 shows selected THz wireless transmission demonstrations at above 300 GHz (including long-distance (>10 m) demonstrations), revealing the relation of data rate, distance, and transmitter THz energy per bit per distance[9,10,14,15,17,18]. One should note that these numbers are only indicative as the measurement conditions in these demonstrations are different in terms of referenced bit error rate (BER) level and the complexity of employed digital signal processing (DSP). More comprehensive comparisons among these demonstrations will need to be performed with common metrology standards, which are yet to be established.

## Results

### Fully integrated on-chip THz transmitter for a THz FDM high-speed WLAN.
Figure 2 shows a conceptual diagram of a fully integrated on-chip photonic-wireless THz transmitter, where a THz frequency division multiplexed (FDM) high-speed wireless local area network (WLAN) is used as an exemplary application. The THz FDM system includes a shared chip-based optical frequency comb (OFC) generator and several integrated THz photonic-wireless transmitters. The centralized OFC can be distributed to each THz transmitter to perform injection locking. Each THz transmitter consists of dual-DFB lasers, a modulator, and a UTC-PD integrated THz antenna. The free-running dual-DFB lasers become phase-coherent after injection-locked to the frequency comb. One laser output is modulated with a broadband signal, whereas the other remains unmodulated as a local oscillator (LO). These two lightwaves are then combined and launched into the UTC-PD for heterodyne photomixing. The generated THz signal is emitted from the THz antenna. The wavelengths of the DFB lasers in each THz transmitter can be tuned to be injection-locked to different frequency comb tones with different spacing, generating different THz carrier frequencies to enable the THz FDM scheme. Such a THz FDM high-speed WLAN can support high-security indoor communications since each user can be allocated a distinct frequency channel.

### Dual-DFB PIC injection-locked to OFCG for high-purity THz generation.
Figure 3(a) shows the monolithically integrated dual-DFB PIC, which consists of two tunable DFB lasers that are coupled to the 3-dB multimode interference (MMI) coupler. The wavelength of each DFB tone is tuned by the thermal tuning current of the integrated heater, and the intensity is controlled by the injection current. RF current injection is supported by the GSG transmission coplanar waveguide (CPW). The superimposed optical dual-DFB signal after the MMI coupler transmits through the spot-size converter (SSC) on the cleaved facet with anti-reflection (AR) coating. The integrated PIN-PD in the PIC

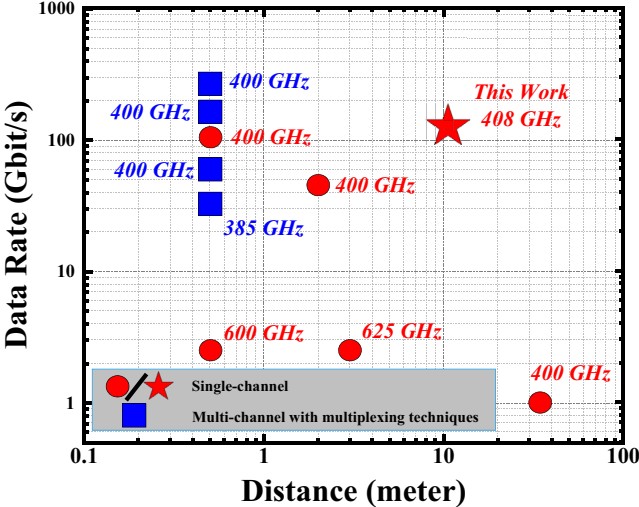

**Fig. 1 State of the art.** A non-exhaustive summary of state-of-the art in photonics-assisted THz transmissions at >350 GHz region with respect to the demonstrated data rates and transmission distances.

**Table 1 Comparison of selected representative THz transmission demonstrations above 300 GHz in terms of normalized transmitter THz energy.**

| Frequency band | Emitted THz power (dBm) | Distance (meter) | Bit rate (Gbits⁻¹) | Measurement conditions | Normalized power (J bit⁻¹ m⁻¹) |
|---|---|---|---|---|---|
| 400 GHz[14] | 10 | 35 | 1 | Real-time @BER $1 \times 10^{-9}$ | $2.9 \times 10^{-13}$ |
| 300 GHz[18] | 0 | 110 | 115 | Offline @BER $1.25 \times 10^{-2}$ | $9 \times 10^{-17}$ |
| 300 GHz[17] | −13 | 100 | 50 | Real-time @BER $9.5 \times 10^{-4}$ | $1.0 \times 10^{-17}$ |
| 300–500 GHz[9] | −24 | 0.5 | 150 | Offline @BER $3.8 \times 10^{-3}$ | $5.3 \times 10^{-17}$ |
| 300–500 GHz[10] | −24 | 0.5 | 260 | Offline @BER $2 \times 10^{-2}$ | $3.1 \times 10^{-17}$ |
| 400 GHz[15] | −24 | 0.5 | 106 | Offline @BER $2 \times 10^{-2}$ | $7.5 \times 10^{-17}$ |
| 400 GHz [This work] | −24 | 10.7 | 131 | Offline @BER $2.7 \times 10^{-2}$ | $2.9 \times 10^{-18}$ |

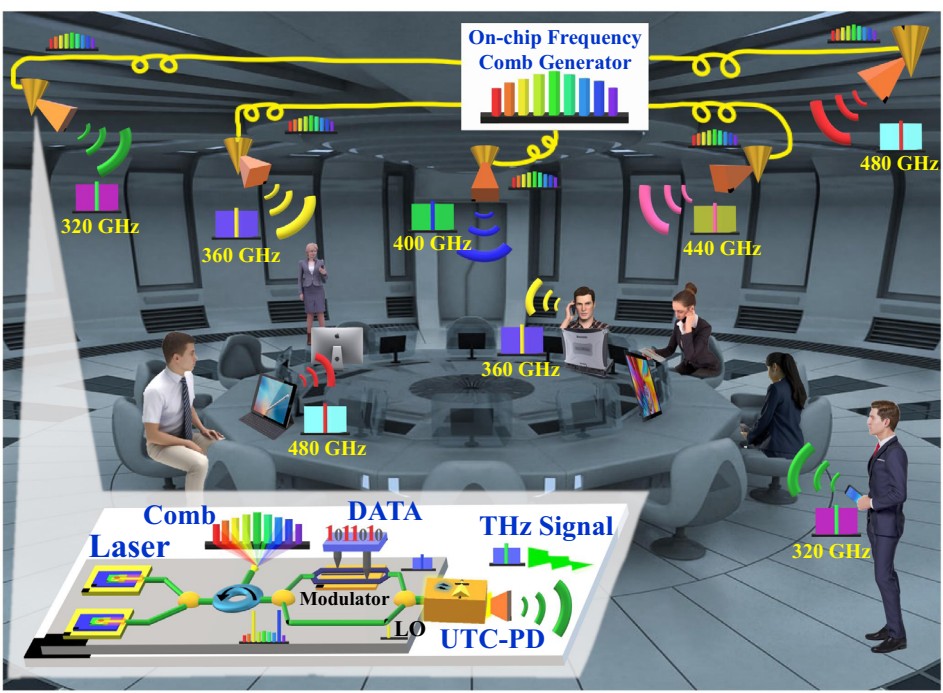

**Fig. 2 A conceptual diagram of a high-speed wireless local area network (WLAN) based on THz frequency division multiplexing (FDM) with fully integrated on-chip THz transmitters injection-locked by a optical frequency comb (OFC).** The shared frequency comb is distributed to each integrated THz transmitter, consisting of dual-DFB lasers, data modulator, UTC-PD, and THz antenna. LO local oscillator; UTC-PD uni-traveling carrier photodiode.

supports up to 40 GHz on-chip optical heterodyning generation. Since the operational ranges of the PIN-PD and the GSG are far below the targeted THz band, they are not studied in detail in this paper. The tuning range of the wavelength spacing between the two free-tuning DFB modes is between 0 nm and 10.7 nm, which corresponds to 0–1.4 THz at 1.5 µm C-band. An off-the-shelf 9.951-GHz MLL-based OFC generator (OFCG) is used to injection-lock the two modes simultaneously to keep them correlated, thus reducing the phase fluctuations between them. In our experimental configuration, the two lasers are set to 1555.575 nm and 1558.975 nm, so to be injection-locked to two selected coherent comb lines from the OFC.

Figure 3a also shows the overall experimental setup for characterizing the OFCG-locked dual-tone laser. An optical bandpass filter (OBPF), a polarization controller (PC), a polarizer, and a polarization-maintaining variable optical attenuator (VOA) are placed between the OFCG-locked dual-DFB PIC and the UTC-PD to align the polarization states and control the optical power of the input signal. The generated THz signal centered at 408 GHz is emitted from the UTC-PD output into the free space. A 10.7-m wireless link is established between a pair of THz lenses with 100-mm diameter and 200-mm focus length, acting as beam collimator. At the receiver, The 408-GHz THz signal is then down-converted to an

intermediate frequency (IF) of 10 GHz with a Schottky Barrier Diode (SBD)-based subharmonic mixer, with an electrical LO at 398 GHz, 12-time upconverted from a fundamental frequency of about 33.17 GHz. An electrical spectrum analyzer (ESA) is used to detect and analyze the down-converted IF signal.

Figure 3b shows the typical behavior of the DFB laser. One can observe that as the injection current raises, the intensity and wavelength continuously increase. The intensity and wavelength can also be further tuned with the heater current. The wavelength tuning range for DFB-1 is 1552–1556 nm, and for DFB-2 is 1556–1560 nm. The optical signal-to-noise ratio (OSNR) is around 60 dB during single-wavelength operation. Figure 3c shows the optical spectra of the OFC and the optically injection-locked two-tone emission monitored at ports 1 and 3 of the optical circulator. From the OFC spectra, one can see that the OSNR of the tone of OFC is ~40–50 dB. However, when both lasers are aligned to the OFC and properly biased to have the 3.3-nm (408-GHz) wavelength separation, the OSNR of the two selected OFC tones increases to 50–60 dB. This observation indicates that the optical injection locking method can increase the OSNR of the tones by ~10 dB in respect to the original OFC. The unwanted adjacent modes can be further suppressed with a sharp optical filter or with a OFC of wider spacing.

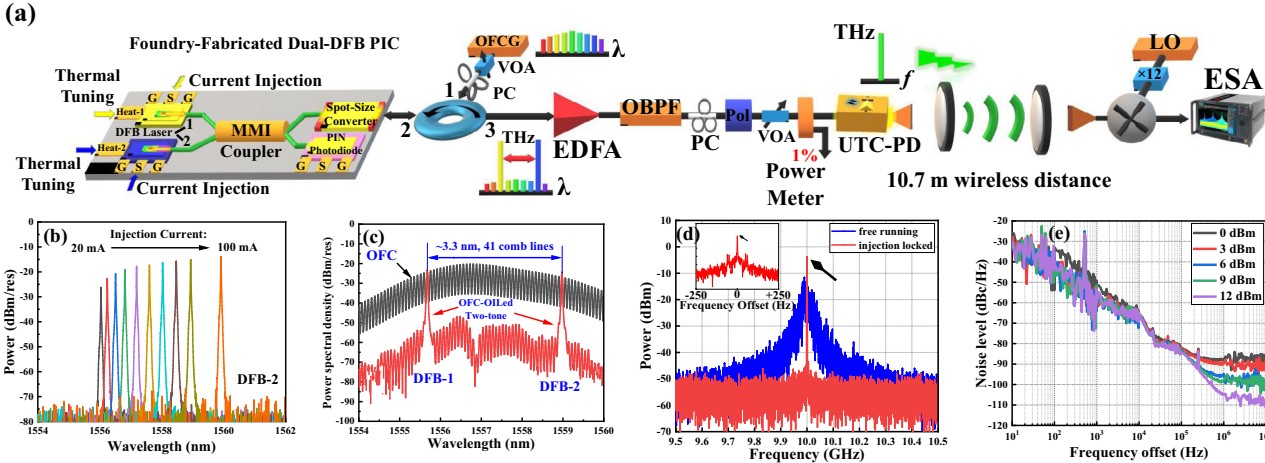

**Fig. 3 Experimental setup and results of high-purity THz carriers generation. a** A master-slave optical injection-locking-based THz carrier generation scheme. The two DFB lasers are injection-locked as slaves by the master OFCG. The polarization state and injection strength are controlled and optimized by a polzarization controller (PC) and a variable optical attenuator (VOA), respectively. The output OFCG-locked two-tone signal passes through the port 3 of the circulator. The wavelength of each DFB tone is tuned by the thermal tuning current of the integrated heater, and the intensity is controlled with the injection current. The two-tone signal with separation of 408 GHz is then amplified, band-pass filtered, and polarization-aligned before being photomixed at the UTC-PD, generating a 408-GHz signal with broadband modulation. The signal is propagated over 10.7-meter distance before being detected and down-converted by a Schottky Barrier Diode (SBD)-based subharmonic mixer. An electrical spectrum analyzer (ESA) is used t monitor the down-converted intermediate frequency (IF) signal. **b** The optical spectra of the DFB-2 laser wavelength shifted between 1556 nm and 1560 nm, driven by injection current changing between 20 mA and 100 mA. The laser intensity varies accordingly. **c** The optical spectra of the two lasers and the OFC. The laser wavelengths are set to 1555.375 nm and 1558.975 nm with 3.3 nm (408 GHz) separation, spanning over 41 comb lines of 9.951 GHz comb-spacing from the MLL-based OFC. **d** The electrical spectra of the IF signal with and without the injection locking. The IF signal drifts around 10 GHz when free-running, shown as the blue trace. After being injection-locked to the OFC, the linewidth of the IF signal is much narrower and the power is 10 dB higher than the free-running case, shown as the red trace. **e** The single-sideband (SSB) phase noise as a function of comb-injection powers. The SSB phase noise decreases at higher injection power, particularly observable from the phase noise floors at >100 kHz frequency offset.

Figure 3d shows the electrical spectra of the IF signal with and without the injection-locking. When the two DFB lasers are free running, the spectrum is shown as the blue trace, whereas when the lasers are properly biased and injection-locked to the OFC, the IF spectrum is shown as the red trace. The injection power of the master comb laser measured with an optical power meter at port-1 of the optical circulator is around 10 dBm. When free-running, the dual-DFB lasers are uncorrelated and generate a high-level phase perturbation, resulting in large linewidth in the sub-GHz scale and a long-term frequency instability. On the contrary, as shown in the inset of Fig. 3d, a Hz-level 3-dB linewidth is achieved after the injection-lockng by the OFC. Figure 3e shows the measured single-sideband (SSB) phase noise power spectral density (PSD) of the down-converted synthesized signal for the dual-DFB lasers separated by 408 GHz (3.3 nm). The impact of injection power level on the phase noise PSD can be clearly observed. A higher injection power level yields lower phase noise floor. At >9 dBm injection power, the phase noise of the heterodyne signal is reduced to the level of $<-100$ dBcHz$^{-1}$ at >1 MHz offset.

**Experimental demonstration of 131 Gbits$^{-1}$ THz wireless transmission over 10.7 m**. Figure 4 shows the experimental setup for high-speed wireless transmission based on the injection-locked dual-DFB PIC THz generator. At the transmitter, we use aforementioned dual-DFB laser chip to generate two continuous waves (CWs) at 1555.675-nm and 1558.975-nm, as shown in Fig. 4(a) inset. The MLL-based OFC with 9.951-GHz-spacing is used to injection-lock the two CW modes, resulting in correlated frequency and phase with high purity and long-term stablity. Again, the spacing between the two DFB lasers is configured at 408 GHz to generate the THz carrier. The two coherent tones generated in the dual-DFB laser chip are separated by a

demultiplexer. One of the tones is modulated with broadband complex signal at an optical in-phase (I) and quadrature (Q) modulator (IQM), and the other tone acts as an optical LO for heterodyne mixing to generate the THz wave.

A two-channel arbitrary waveform generator (AWG) with 64-GSa per second sampling rate is used to generate the IQ-OFDM signal. The length of the inverse fast Fourier transform (IFFT) of the IQ-OFDM signal is set to 1024 and the length of cyclic prefix (CP) is 16. The first subcarrier is set to null. The OFDM symbols are mapped from a random binary sequence generated from MATLAB based on Mersenne Twister with shuffled seed numbers. The generated optical 16-QAM-OFDM signals are firstly amplified by an erbium-doped fiber amplifier (EDFA), before being filtered by an OBPF to remove the outband amplified spontaneous emission (ASE) noise. The power of the optical signal is adjusted by a VOA before combining the optical LO so that power ratio between the optical LO and signal is kept balanced to achieve the highest photo-mixing efficiency at the UTC-PD[19].

The baseband modulated signal and the optical LO are aligned in polarization, power combined and launched into the broad-band UTC-PD. We use a polarization-maintaining VOA to control the UTC-PD input power. The inset of Fig. 4b shows the optical spectra of the combined signal and LO. A THz signal with carrier frequency centered at 408 GHz is generated and emitted from the UTC-PD output into a 10.7-m line-of-sight (LOS) wireless link. A photo of the link setup is shown in Fig. 4d. It is noted that 408 GHz carrier frequency was chosen after an optimization process, as the THz receiver shows the optimal performance ~400 GHz. The same aforementioned pair of THz lenses are used to collimate the THz beam. At the receiver, the THz signal is down-converted to an IF by employing the SBD-based sub-harmonic mixer operating in the 0.3–0.5 THz band,

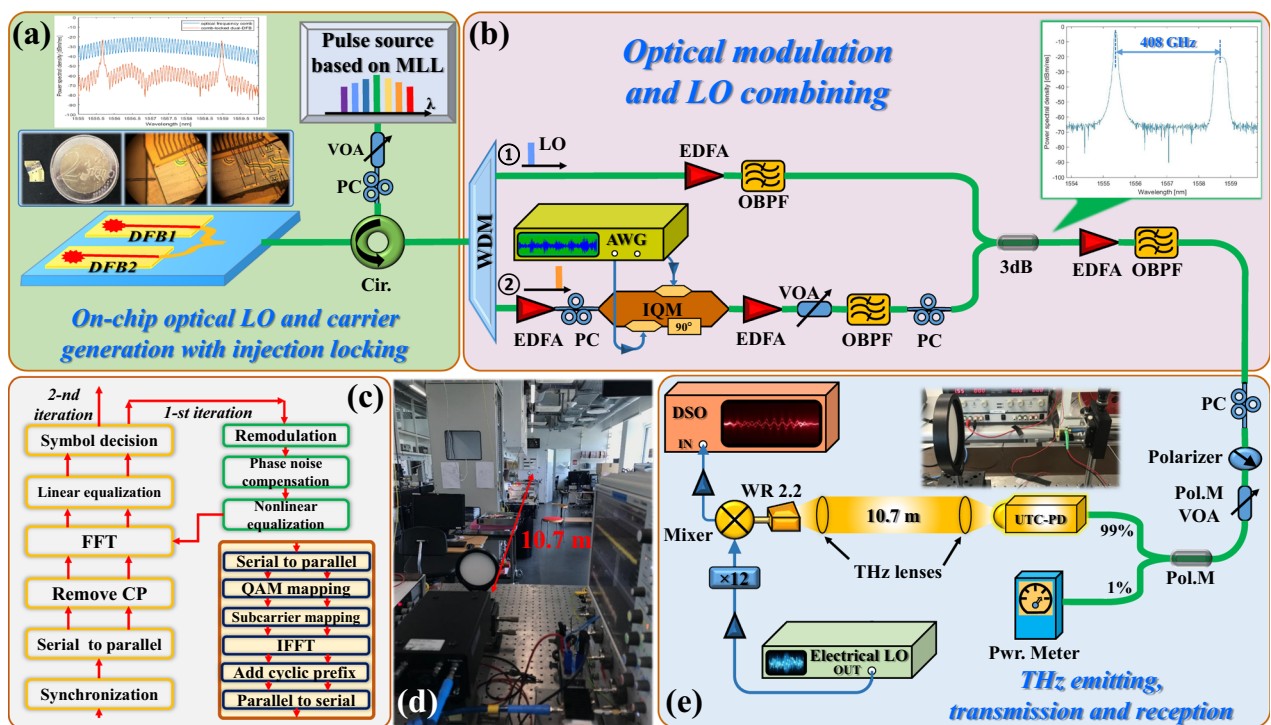

**Fig. 4 Configuration of high-speed THz wireless transmission.** Experimental setup of the single-channel photonic-wireless transmission system carrying 131.21 Gbits-1 16-QAM-OFDM signal over a 10.7-m distance: **a** the injection-locked dual-DFB PIC for optical LO and carrier generation; **b** the broadband signal modulation and recombining with the LO; **c** the DSP routine for signal equalization and demodulation; **d** a photo of the THz link setup and **e** THz emission, transmission, and reception.

driven by a 12-time (×12) frequency multiplied electrical LO. In this configuration, the fundamental frequency of the electrical LO is set to be 32 GHz, resulting in a corresponding IF signal centered at 24 GHz. We used an RF amplifier with 45 GHz bandwidth to firstly amplify the IF signal before capturing it with a 160 GSas⁻¹ real-time digital sampling oscilloscope (DSO) with 63 GHz analog bandwidth. The converted digital samples are stored, processed, and analyzed offline.

Figure 4s shows the structure of the offline DSP routine for channel equalization and signal demodulation. The channel equalization consists firstly a linear equalization (LE) stage, a phase noise compensation (PNC), and finally a nonlinear equalization (NLE) stage. In the first stage, a pilot-based one-tap LE module is employed to compensate for the system linear response. After that, a simple least-squares method-based PNC is applied to reduce the impairment from the common phase error[20,21] dominated phase noise after LE, owing to the low carrier linewidth. Finally, a Volterra series-based nonlinear model, considering the 2nd-order and the 3rd-order distortion terms, is used for nonlinear impairment estimation and conpensation (see details in Methods section and Supplementary information). In our system, the nonlinear distortions mainly come from the saturation effects of optical IQ-MZM[22], saturation effects of broadband electrical IF amplifier[23], and the high-order harmonics from the mixer[24]. All these nonlinear impairments will introduce inter-carrier interference to the OFDM signal transmission and deteriorate the EVM performance. Thanks to the injection locking scheme in our proposed system, the laser chirp effect is significantly suppressed, and the nonlinear impairments from the lasers are reduced.

We evaluate the transmission performance of the single-channel THz system over the 10.7-m link distance. Figure 5a shows the BER performance as a function of the launched optical power into the UTC-PD for four-channel equalization cases (w/o equalization, LE, LE + PNC, LE + PNC + NLE). One can observe that by employing the nonlinear DSP (LE + PNC + NLE) routine, we can successfully achieve a BER performance below the low-density parity-check convolutional codes (LDPC-CC) forward error correction (FEC) threshold (2.7 e-2, 20%-OH, the pre-FEC BER was calculated from the given Q factor in dB as (1/2)erfc(10^{5.7dB/20}/√2))[25,26]. The 16-QAM-OFDM signal has a overall bandwidth of 44.43 GHz, corresponding to a gross bit rate of 157.46 Gbits⁻¹ after subtracting the pilot overhead. Furthermore, after subtracting the FEC overhead, we obtain a net data rate of 131.21 Gbit/s. We also calculated the capacity from the generalized mutual information (GMI)[27], which is 134.56 Gbits⁻¹ at 14 dBm optical power and has ~2.5% variation with post-FEC capacity. Figure 5b–e shows the corresponding signal constellation diagrams before and after each channel equalization stage at 14 dBm optical power. Figure 5f, g shows the electrical spectra of the 44.43 GHz OFDM signal before and after the down-conversion from the IF to the baseband. In this experiment, the overall system performance is ultimately limited by the received signal SNR. A mean SNR of 13.45 dB is achieved, as shown in Fig. 5g.

## Discussion

We have experimentally demonstrated a single-channel 131.21 Gbits⁻¹ net rate THz-band wireless transmission over a 10.7-m distance, employing 16-QAM-OFDM modulation, nonlinear DSP flow, and OFC injection-locked heterodyne THz generator based on generic foundry fabricated PIC. Other components in the system, such as modulator, demultiplexer, circulator, VOA, and optical filters, also have integration solutions[28]. Therefore, the scheme of using a monolithic dual-DFB laser chip THz generation shows great potential for fully integrated THz transmitters and further a cost-effective and energy-efficient THz FDM high-speed WLAN.

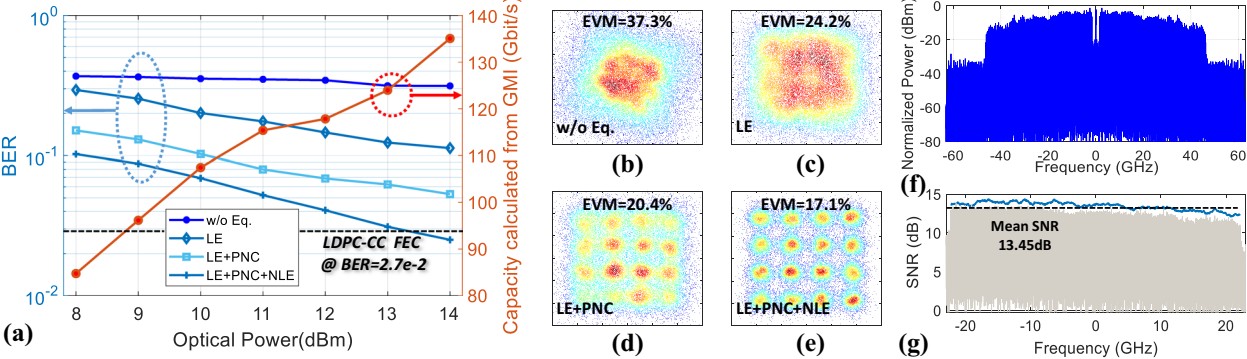

**Fig. 5 Transmission results. a** BER versus the optical power launched into the UTC-PD for 4 cases with different combinations of DSP modules, the BER y-left-axis is labeled in logarithm-scale with light-blue color grids, the y-right-axis is labeled in linear-scale. **b–e** The signal constellations before and after the three channel equalization stages at 14 dBm optical power (color in logarithm-scale). **f** The electrical spectrum of the IF signal before digital down conversion to baseband. **g** The SNR (blue curve) versus the frequency after down-conversion, the gray-color shading is the shape of the electrical spectrum of the 16-QAM-OFDM signal after down conversion.

It should be noted that compared with the method of selecting two lightwaves directly from an optical frequency comb in our previously reported work[15], the presented optical injection locking method provides high gain with low added intensity noise[29]. Here, it can be seen in Fig. 3c that the optical injection locking method can increase the OSNR of the tones by ~10 dB in respect to the original comb. In addition, the injection locking scheme can save an optical filter, which can reduce the system's cost and complexity. It is worth noting that the adopted comb-based optical injection locking scheme in this work is only one of the possible approaches for optical phase-locking to achieve frequency stable and phase-locked optical tones from the dual-DFB PIC. Other on-chip optical phase-locked methods can also be used for a stable beating, such as monolithically integrated optical phase-locked loop (OPLL) PIC[30]. Additionally, though challenging and not yet demonstrated, it is not technically prohibitive to further integrate the frequency comb with the dual-DFB PIC, even monolithically. Several chip-based optical frequency combs have been reported, such as on-chip mode-locked laser (MLL) based on generic InP and III-V-on-silicon photonic integration platforms[31], on-chip mode-locked Kerr combs based on the $Si_3N_4$ micro-resonator[32], and III-V-on-silicon-nitride MLL chip[33]. Therefore, one can expect a potential fully integrated PIC with either the OPLL- or the frequency comb-based scheme to support the compacity advantage of the PIC approach. Due to the limitation of the experimental condition, we used an off-the-shelf MLL for comb generation in this experiment for proof-of-concept purposes. In our future research, we aim for a monolithically integrated dual-DFB PIC with effective phase-locking.

The main limitations of the communication system are the noise level of the THz receiver and the low emitted power of the UTC-PD. In the future, to make this approach a reality for practical applications, the perspectives would include on-PIC chip optical frequency comb generation, integrated multiple lasers chip, integrated optical modulator, couplers, and filters, integrated THz transmitter with more output power, highly directional THz beamforming and beam steering[2,34]. In detail, PIC chip-based optical frequency combs can be realized, such as on-chip MLLs[31–33]. Integrated multiple lasers chip has been achieved on the III-V PIC platform[6]. Other optical components also have integration solutions[28]. The photonics-based THz phased array supporting more output power, THz beamforming, and beam steering has been developed towards monolithic integration[35].

As effective DSP and channel coding are already widely deployed in most commercial systems for both wireless and optical communications, we use an advanced DSP routine and benchmark our transmission performance with soft-decision LDPC-CC FEC threshold in this demonstration. However, one should note that the drawbacks of employing such sophisticated DSP and FEC decoder are more significant receiver power consumption and higher system latency, compared with the systems operating in 'real-time' with virtually error-free performance. Especially, the highly demanding channel coding and decoding dominate the baseband computations. In the experiment, the LDPC-CC FEC limit uses 20% coding overhead and contributes most of the power consumption and latency in practice. Therefore, for specific high-end applications where low latency is critical, one should consider using low-complexity signal processing and light FEC variants at the cost of lowering the system data rate. It is also worth noticing that considering the baseband signal processing, incorporating application-specific integrated circuits (ASIC) in a holistic framework is vital for decreasing the time-to-market of the THz communication system. A joint optimization between algorithms (especially coding scheme) and ASIC architecture across all blocks is expected to optimize the throughput, power consumption, and latency.

Finally, it is worth noting that such photonics-based THz technologies are still in their initial research phase. In the long-term, transceiver technologies based on all-electronic approaches, including the III-V semiconductor and silicon-based CMOS technologies, should all be included in performance comparisons with established common metrology standards upon the maturity of the adopted photonics-based THz technologies.

## Methods

**Generic foundry approach for photonic integrated circuit**. Open-access generic photonic integration technology enables building active and passive components on a single substrate without the need for additional high-precision assembly. A wide variety of functionalities are feasible by using standard building blocks (BBs) including, but not limited to lasers, amplifiers, photodetectors, phase modulators, filters, wavelength multiplexers, power splitters, couplers, and combiners. Mainly, they can be composed of a combination of waveguides of different widths and lengths. BBs are predefined to ease the design flow and can be called and parameterized according to the foundry's process design kit (PDK). The photonic integrated circuits (PICs) are manufactured within a multi-project wafer (MPW) service whereby mask and wafer area are shared, and so is fabrication cost among multiple different users. The PIC used for the THz communication system was developed in such an MPW run, by Fraunhofer HHI, JePPIX (Joint European Platform for Photonic Integration of Components and Circuits).

**Digital signal processing**. The DSP flow at the transmitter is similar to a conventional OFDM system, shown in the inset of Fig. 5. A mapping module is used to transform the parallel binary data in different subcarriers to QAM symbols. After mapping, the inverse fast Fourier transform (IFFT) module transforms the frequency domain symbols into the time domain to realize the OFDM modulation.

Then, a cyclic prefix (CP) is inserted to reduce the influence of inter-symbol interference (ISI). To capture the OFDM data frame in the scope, we add a pseudo-noise (PN) sequence with a length of $2^7$-1 at the head of serial OFDM data for synchronization. At the receiver, the signal is first captured from the scope, and then we use the autocorrelation of pseudo-noise (PN) sequence to find the frame head in the captured frame. After synchronization, the CP is first removed to mitigate the influence of multi-path interference induced ISI. Then, the fast Fourier transform (FFT) module transforms the time domain symbols into the frequency domain to realize the OFDM demodulation. After that, equalization is used to recover back the signal. The detailed derivations of the DSP routine and optimization of DSP parameters are described in the Supplementary Information.

**Experiment**. The experimental setup is shown in Fig. 4. The coherent OFCG is based on an off-the-shelf MLL with 9.951-GHz mode spacing. The optical power of the OFC injected into the dual-DFB chip is around 10 dBm. The gain of the EDFA in the path of optical LO after the demultiplexer is tunable to keep the power balance between optical LO and modulated signal. The coupled optical LO and modulated signal are launched into the UTC-PD, where the input power is up to 14 dBm controlled by a VOA. The THz output power of the UTC-PD with the conversion efficiency of 0.15 A/W is from −30 dBm to −20 dBm, depending on the optical input power. Then the Schottky mixer-based THz receiver with extremely high sensitivity down-converts the single-channel THz signal to IF. The IF output is fed into the DSO for analog-to-digital conversion. Finally, the digital signals are processed and analyzed offline.

For the THz transmitter in this system, we employed a UTC-PD as the photomixer. UTC-PD has an ultrafast response with a relatively high output power in the THz range compared with other photomixers. The response of the UTC-PD is determined by the electron transport in the whole structure, where the electrons exhibit velocity overshoot ($3-5 \times 10^7$ cm s$^{-1}$) in the carrier collection layer[19,36]. In addition, UTC-PDs generate higher output saturation current due to the reduced space charge effect in the depletion layer, resulting from the high electron velocity in the depletion layer[36]. For our experiment, the UTC-PD is a commercial device (IOD-PMAN-12001) manufactured by NTT. It's an antenna-integrated photomixer integrating a hybrid-absorber UTC-PD and a bow-tie antenna. It operates in C-band with maximum input optical power of 15 dBm. The typical DC responsivity is ~0.15 AW$^{-1}$, and it covers THz output frequency range of 200–1500 GHz per the manufacture's specifications. With our experimental configuration, the emitted THz signal power from the UTC-PD is found to be ~−24 dBm. For the THz signal reception, a commercial even harmonic mixer from Virginia Diodes (VDI, WR2.2MixAMC) is used. It supports input THz ranging from 325 GHz to 500 GHz with an IF output bandwidth of ~40 GHz. The sensitivity of the THz input is <−30 dBm, and the conversion loss is ~17 dB.

## Data availability

The measurements data generated in this study have been deposited in https://doi.org/10.5281/zenodo.5855331.

## Code availability

All codes of the DSP algorithms used in this study are embedded in a larger framework, which, together with specific user instructions can be available from the corresponding authors upon reasonable request.

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

## Acknowledgements

This work was supported in part by the EU H2020 Marie Sklodowska-Curie grant agreement no. 713683 (COFUNDfellowsDTU) (S.J.); the EU H2020 Marie Skłodowska-Curie Grant agreement no. 642355 FiWiN5G, 713964 MULTIPLY, and 101032236 COINCOST (M.C.L.); the Danish center of excellence CoE SPOC under Grant DNRF123, the Villum young investigator program grant of 2MAC and the Independent

Research Fund Denmark under the grant of 9041-00395B (S.J., H.H., and L.K.O.); the Swedish Research Council (VR) projects 2019-05197 (X.P.) and 2016-04510 (O.O.); National Natural Science Foundation of China (61775162, 61331010, 61722108, 61775137, 61671212, 61771424, and 62101483), the National Key Research and Development Program of China (2020YFB1805700) (L.Z. and X.Y.); TERAWAY, funded by the European Union's Horizon 2020 under G.A No 871668 and is an initiative of the 5G PPP. MARTINLARA, funded by Comunidad de Madrid, S2018/NMT-4333. European Space Agency Contract No. 4000135351/21/NL/GLC/my (R.G. and G.C.).

## Author contributions

S.J. and H.H. proposed the THz photonic-wireless transmission based on the monolithically integrated dual-DFB lasers chip injection-locked by a coherent OFC. S.J. designed the overall experiment. M.C.L. developed the monolithically integrated dual-DFB lasers chip. L.Z. developed the OFDM-16-QAM modulation and DSP routine. S.J., M.C.L., L.Z., O.O., and A.U. carried out the experiment. D.K., X.P., L.K.O., and H.H. assisted in discussing and interpreting the results. S.J. wrote the draft of the manuscript. X.P., X.Y., S.X., S.P., J.C., G.C., T.M., L.K.O., and H.H. edited the manuscript. H.H. coordinated and supervised the experiment. All the authors discussed the results and reviewed the manuscript.

## Competing interests

The authors declare no competing interests.
