## [Peer Review File · Nature Communications]

REVIEWER COMMENTS

Reviewer #1 (Remarks to the Author):

This paper shows a feasibility of over 100 Gbit/s wireless transmission on 0.4 THz carrier with a distance of 10.7 m, which seems to be a record performance of an indoor application of the THz-wave transmission. The demonstrated results are very informative and interesting for readers in a field of a large capacity indoor wireless transmission. However, unfortunately, an advantage of the presented configuration and some experimental conditions are unclear.

The paper is better to be resubmitted after making clear the above matters.

For example, the followings should be made clear.

An information of the bit rate of carrier is not described. Thus, it is not clear how to estimate the single-channel net rate of 131 Gbit/s.

For making a pair of lightwaves, another possible way is to select two lightwaves directly from an optical frequency comb. What is an advantage of the presented method compared with the direct selection?

A narrow linewidth of the lightwaves is obtained and it enables a low noise signal. The impact on the bit error rate is not clear. Can a relation between the linewidth and the bit error rate be discussed?

Reviewer #2 (Remarks to the Author):

The paper demonstrates the use of PICs towards generation of dual optical lines for THz generation. Then, the transmission of 0.4 THz channels is evaluated. The paper combines different approaches for Tx (PIC chip) and post-processing steps (for Rx) somehow already discussed in the literature, by authors. This contribution, while being a combination of many devices/systems sounds more incremental to me than completely new approach.

In addition, I feel that several details are missing, for example about the THz devices, for people to be able for full comparisons with other similar works in the field (see my comments).

I have a couple of questions/comments.

The main advantage of the PIC approach as highlighted by authors is the compacity. However, considering just the PIC, the DFB lasers are not locked and a comb frequency is required, thus compacity is more limited.

Q. How would operate the system with only the PIC chip (without any comb) ?

It would have been interesting to describe more on the the UTC-PD used, the power level and I would have expected to see a link budget to be able to compare with other approaches.

Q. System-level figures should be included and deeper description of THz Tx and Rx (which I assume to be commercial ones ?)

Concerning the overall compacity of the THz system, the approach has to be compared with CMOS or InP/GaAs systems, which has a moderate (InP/GaAs) or high (CMOS) integration level demonstrated also 100 Gbit/s.

Q. How is the impact of the proposed scheme over other works ?

It is claimed that the 131 Gbit/s over 10.7 m is a record in terms of THz power/bitrate/distance. The authors reported 160 Gbit/s over 0.5 m with I think less antenna gains. However, the single-data-rate performance that is often used in THz communications however it is important to compare the energy consumption of the overall system and the system performance like the real-time bit-error-rate, the latency time, ...

Most of the THz systems make use of signal processing at receiver stage, thus this implies energy consumption. For fig. 1b., I think only the THz transmitter is considered. Thus, if the receiver makes use of complex post-processing steps, it induces some distortion when comparing different systems.

As a consequence, the figure 1.a and b suggests that the proposed scheme is the highest reported performance, but other parameters should be integrated for a global comparison with other

approaches/systems. In my opinion, a deeper comparison between systems should be considered, including the receiver and post-processing.

Remark : usually, the figure of merit of any communication system is the energy per bit (pJ/bit) required for a specified performance (BER for example), rather than pJ/bit/m.

As an exemple, the reference [16] reported a real-time performance, which means no latency time nor post-detection using signal processing. Here, the post-processing is described in the text and contains several steps. This should be considered for fair comparison with other systems as a real-time performance is more appropriate for practical use-cases.

I feel that this is thus not completely fair to compare on the same graph (fig.1b) the work of [16] that has a real-time performance (BER < 10⁻⁹) and the proposed approach that comes with BER > 10⁻², because it could lead for the readers to biased conclusions. THz communications is still requiring common metrology standards, that not yet exists, thus the community has to take care of system-comparisons.

Q. A more comprehensive/deeper comparison should be considered, by including global energy consumption (because at the end we need a full THz system, thus Tx and Rx consumption should be integrated).

Somehow the receiver consumption, receiver sensitivity has to be described.

Q. Authors's previous work which seems to use the same type of UTC-PD should be included as well in this comparison (fig 1.b for example) to evaluate the performance increase of the proposed work.

Remark : speaking about a « record » has to come with full analysis with other systems in my opinion.

In the fig. 5, from (d) to (e), the NLE seems to enhance the constellation but the non-linear impairments of fig.5. (d) already seems limited. I wonder in which extent the NLE is efficient, of this is due to the used color scale for (b),(c),(d),(e).

Q. The authors should detail the color-scale, is this linear or not ? And qualify the improvement in terms of EVM (%) of the constellations ?

The overall BER values obtained are high and higher than 10⁻², even with complex post-processing, which means that practical implementation.

Q. What is the main limiting factor in this system and what would be the perspectives to make this approach a reality for use-case of fig. 2 ?

Reviewer #3 (Remarks to the Author):

This manuscript describes a photonic integrated circuit coupled to an optical frequency comb, which produces two high-purity tones from two injection-seeded DFB lasers. These are then mixed to produce a narrowband tone near 400 GHz. One of the optical tones can be modulated first, thus producing a high-data-rate modulation on the THz signal. The work represents a significant step in the demonstration of photonics-based wireless links in the THz regime. The achieved results are impressive, and will attract a great deal of attention from the community. Therefore I recommend publication, after the authors consider a couple of small (but important) modifications.

1. The abstract suggests that the result at 10m range was achieved with a rather low THz output power (-24 dBm?). This is a bit surprising, so it would be valuable if the authors included an analysis of the link budget (this could possibly be put in the supplementary materials, if there is no room in the main text). What's the free-space path loss at this frequency? What are your antenna gains? In light of that link budget analysis, does it make sense that you are able to achieve ~13 dB of SNR? Well, but that SNR is after all of the DSP, so then what should the link budget be compared to? These considerations are important for understanding how to extrapolate this system demonstration to, e.g., longer range.

2. It is clear from Fig. 5 that the nonlinear equalization and phase noise compensation steps are critical to obtaining a good BER result. What is less clear is why that's so. There is a nice detailed discussion of the signal processing approach in the supplementary materials; but, it would be very nice if the authors could include in the main text some brief discussion of why these steps make such a big difference. Given the Hz-level linewidth and the SSB noise data presented in Fig. 3, I would not have expected a great deal of phase noise, and therefore would not have expected phase noise compensation to have made such a big difference in the constellation diagram. So, can one easily understand why this is so critical? Also, the nonlinearity that gives rise to the requirement for nonlinear equalization presumably originates mostly from the UTC-PD,

correct? Or is there some other important source of nonlinearity here? In the interests of improving the accessibility of the manuscript to a broad audience, these issues deserve some discussion.

3. Just out of curiosity: why 408GHz? Looks like you could have chosen just about any frequency you wanted, given the span of your OFC. Wouldn't you have gotten better performance at, e.g., 340 GHz, with lower FSPL?

We thank the reviewers for their valuable comments to improve the presentation quality.

We have correspondingly responded to all the comments. Please see the details of the point-by-point response to these comments below. All the corresponding modifications to the original manuscript are highlighted in the revised version.

Shi Jia (on behalf of all co-authors)

REVIEWER COMMENTS

Reviewer #1 (Remarks to the Author):

This paper shows a feasibility of over 100 Gbit/s wireless transmission on 0.4 THz carrier with a distance of 10.7 m, which seems to be a record performance of an indoor application of the THz-wave transmission.

The demonstrated results are very informative and interesting for readers in a field of a large capacity indoor wireless transmission. However, unfortunately, an advantage of the presented configuration and some experimental conditions are unclear.

The paper is better to be resubmitted after making clear the above matters. For example, the followings should be made clear.

Q1: An information of the bit rate of carrier is not described. Thus, it is not clear how to estimate the single-channel net rate of 131 Gbit/s.

Authors' response: Thanks for the comments. In the experiment, we generate a 16-QAM-OFDM signal of bandwidth of 44.43 GHz. The gross rate of the signal is the production of the signal bandwidth and its spectrum efficiency, which equals 177.72 Gbit/s (44.43 GHz×4 bits/Hz). The cyclic prefix (CP), linear equalization (LE) pilot, and the FEC-coding redundancy are the main system overhead. The length of the CP and IFFT are 16 and 1024, respectively, which corresponds to the overhead of $16/(16+1024)\approx 1.538\%$. The overhead of the LE pilot is 10%. Besides, the pseudo-code-based synchronization frame also occupies a negligible part of the transmitted frame since it only has 127 samples and has around 0.1% overhead. Moreover, phase noise compensation (PNC) and nonlinear equalization (NLE) only need to be updated in the training part. Therefore, the system line rate is calculated without considering the FEC overhead, which equals 157.46 Gbit/s ($177.72 \text{ Gbit/s} \times (1-1.538\%) \times (1-10\%) \times (1-0.1\%)$). Subsequently, the

system transmission performance reaches the threshold of LDPC-CC FEC with 20% overhead ($2.7E-2$) [1], and the system net rate is calculated extracting the FEC overhead, which equals 131.21 Gbit/s ($157.46 \text{ Gbit/s}/(1+20\%)$). As a cross-verification, we also performed system data rate estimation by calculating the generalized mutual information (GMI) [2], and the estimated net rate equals 134.56 Gbit/s, which has around 2.5% variation with the calculated one. In the revised manuscript, we have added the details of the calculation process of the system net rate.

[1] Agrell, E. & Secondini, M. Information-theoretic tools for optical communications engineers. *In IEEE Photonics Conference (IPC 2018)*, Invited Tutorial 1-5 (IEEE, Reston, VA, 2018).

[2] Alvarado, A., Agrell, E., Lavery, D., Maher, R. & Bayvel, P. Replacing the soft-decision FEC limit paradigm in the design of optical communication systems. *J. Lightw. Technol.* **33**, 4338–4352 (2015).

Revision: Page 29, Supplementary Information Part

3) The details of the calculation process of the system net rate

In the experiment, the bandwidth of the 16-QAM-OFDM signal is 44.43 GHz. The gross rate of the system is calculated by the production of the signal bandwidth and the spectrum efficiency, which equals $177.72 \text{ Gbits}^{-1}$ ($44.43 \text{ GHz} \times 4 \text{ bits} \cdot \text{Hz}^{-1}$). The CP, the LE pilot, and the FEC coding occupy the main part of the system overhead in the DSP routine, whereas the PNC and the NLE only need to be updated in the training stage. The length of the CP and IFFT are 16 and 1024, respectively, which corresponds to the overhead of $16/(16+1024) \approx 1.538\%$. The overhead of the LE pilot is 10%. Besides, the pseudo-code-based synchronization frame also occupies a negligible part of the transmitted frame since it only has 127 samples and has around 0.1% overhead. First, the system line rate is calculated without considering the FEC overhead, which equals 157.46 Gbit/s ($177.72 \text{ Gbits}^{-1} \times (1-1.538\%) \times (1-10\%) \times (1-0.1\%)$). Then, the system transmission performance reaches the LDPC-CC FEC threshold ($2.7E-2$), which occupies 20% overhead, and the system net rate is calculated extracting the FEC overhead, which equals $131.21 \text{ Gbits}^{-1}$ ($157.46 \text{ Gbit/s}/(1+20\%)$).

Q2: For making a pair of lightwaves, another possible way is to select two lightwaves directly from an optical frequency comb. What is an advantage of the presented method compared with the direct selection?

Authors' response: We thank the Reviewer for this comment.

Indeed, for making a pair of lightwaves, another way is to directly select two lightwaves from an optical frequency comb. Conventionally, this is realized using optical filters, for example, a wavelength selective switch (WSS) as our previously reported work [1]. However, this method will usually introduce an insertion loss of 3-5 dB. Then the selected tones need to be subsequently amplified via an EDFA to boost the power. This 'filter + amplifier' scheme unavoidably reduces the optical signal-to-noise ratio (OSNR) and will occupy a large space even on an integrated photonic chip.

However, in our presented method, optical injection locking, as an alternative method, can provide merits as low noise and high gain filtering simultaneously [2]. When seeding the dual-wavelength slave DFB laser with the comb signal, the two carrier frequencies of the slave laser can be tuned to the wavelengths of the comb signal to be extracted. When the slave dual-DFB laser is injection-locked to the selected input tones, it performs optical filtering by amplifying the selected tones but attenuating the out-of-locking-range signals (other tones of the comb). Compared to the 'filter + amplifier' scheme, the presented optical injection locking method provides high gain with low added intensity noise [2]. This manuscript shows in Fig. 3(c) that the optical injection locking method can increase the OSNR of the tones by around 10 dB in respect to the original comb. In addition, the injection locking scheme can save an optical filter, such as WSS, which can reduce system cost and complexity.

On the other hand, a monolithic dual-DFB laser chip is used as the light source in this experiment, and these two lasers are both free-running. For generating the beating wireless carrier with high-frequency stability, the two DFB lasers on the chip need to be phase-locked. The scheme of optical injection locking using a comb provides a possible optical phase-locked method to phase-lock the two free-running DFB lasers. Other optical phase-locked methods can also be used for stable beating.

[1] Jia, S., Pang, X., Ozolins, O., Yu, X., Hu, H., Yu, J., Guan, P., Ros, F. D., Popov, S., Jacobsen, G., Galili, M., Morioka, T., Zibar, D. & Oxenløwe, L. K. 0.4 THz photonic-wireless link with 106 Gb/s single channel bitrate. *J. Lightw. Technol.* **36**, 610-616 (2018).

[2] Liu, Z. & Slavík, R. Optical injection locking: from principle to applications. *J. Lightw. Technol.* **38**, 43-58 (2020).

Revision: Page 13, 2nd paragraph, the corresponding modification in the revised version is as follows:

It should be noted that compared with the method of selecting two lightwaves directly from an optical frequency comb in our previously reported work [15], the presented optical injection locking method provides high gain with low added intensity noise [29]. Here, it can be seen in Fig. 3(c) that the optical injection locking method can increase the OSNR of the tones by around 10 dB in respect to the original comb. In addition, the injection locking scheme can save an optical filter, which can reduce the system's cost and complexity.

References

[29] Liu, Z. & Slavík, R. Optical injection locking: from principle to applications. *J. Lightw. Technol.* **38**, 43-58 (2020).

Q3: A narrow linewidth of the lightwaves is obtained and it enables a low noise signal. The impact on the bit error rate is not clear. Can a relation between the linewidth and the bit error rate be discussed?

Authors' response: Thanks for the comments. The relation between the linewidth and the BER mainly depends on the phase noise's effect on the OFDM transmission performance. Phase noise $\phi(n)$ in the system can be described as a continuous Brownian motion process with zero mean and variance σ_ϕ^2 , which, from a spectral point of view, can be presented as a finite-power Wiener process [1]. The value of the variance σ_ϕ^2 can be calculated as the integration of the phase noise spectrum $S(f)$ [2], which is:

$$\sigma_\phi^2 = \int S(f) df$$

Here, we use the phase noise spectrum under 12 dBm comb-injection power, shown in the following figure.

Then, the phase noise variance σ_ϕ^2 is calculated to be 0.09503 rad^2 . According to [3], the signal-to-noise ratio (SNR) degradation degrSNR follows the expression:

$$\text{degrSNR} = 10 \cdot \log(1 + \sigma_\phi^2 \cdot \text{SNR}) \text{ dB}$$

Therefore, the SNR degradation for $\sigma_\phi^2 = 0.09503 \text{ rad}^2$ is shown in the following figure:

It can be seen that OFDM is quite sensitive to phase noise since the phase noise causes leakage of FFT and destroys the orthogonality among subcarrier signals. What's more, the influence can be categorized as common phase error (CPE) and inter-carrier interference (ICI) [4]. Assuming the OFDM symbol is:

$$x(n) = \sum_{k=0}^{N-1} s_k \cdot e^{j(2\pi/N)kn}, \quad n = 0, 1, \dots, N-1$$

where s_k is the QAM signal, N is the number of subcarriers. Since the phase noise $\phi(n)$ is small, the phase noise effect can be approximately expressed as:

$$e^{j\phi(n)} \approx 1 + j\phi(n)$$

In this case, the demultiplexed signal at the receiver side is:

$$y(k) \approx s_k + \frac{j}{N} \sum_{r=0}^{N-1} s_r \sum_{n=0}^{N-1} \phi(n) \cdot e^{j(2\pi/N)(r-k)n} \approx s_k + c_k$$

The term c_k is the effect of phase noise on the OFDM signal. When $r=k$, c_k is the CPE interference, we have a common error added to every subcarrier by a rotation of the constellation. Thanks to the common error nature of CPE, it can be easily compensated with the least-square PNC method. Comparatively, when $r \neq k$, c_k is the ICI interference, which is the summation of the information of the other $N-1$ subcarriers, each multiplied by a set of complex numbers. Because of the random and nonlinear nature of ICI, its compensation is complicated, and performance is still getting influenced.

In our scheme, the phase noise bandwidth is limited since the linewidth of the carrier is narrow, so the CPE dominates over ICI in our scheme. Thus, a simple PNC scheme could compensate for the phase noise effect of the OFDM frames and reduce the performance degradation, which can be observed in the constellation graphs in the paper (Fig. 5). What's more, the NLE could further compensate for the ICI-induced nonlinear impairments and improve the performance.

In the revised manuscript, we have added more reference papers to give the readers more information about the effect of linewidth on the system performance.

- [1] Wu, S. & Bar-Ness, Y. OFDM systems in the presence of phase noise: consequences and solutions. *IEEE Transactions on Communications* **52**, 1988-1996. (2004).
- [2] Robins, W. P. Phase noise in signal sources: theory and applications. **9**, IET. (1984).
- [3] Moeneclaey, M. The effect of synchronization errors on the performance of orthogonal frequency-division multiplexed (OFDM) systems. *In Proc. COST 254'Emerging techniques for Communication Terminals'*. (Toulouse, France, 1997).
- [4] Armada, A. G. Understanding the effects of phase noise in orthogonal frequency division multiplexing (OFDM). *IEEE transactions on broadcasting*. **47**, 153-159. (2001).

Revision: Page 11, 2nd paragraph, the corresponding modification in the revised version is as follows:

... Since the carrier linewidth is low, the signal is equalized with a least-squares method-based PNC to reduce the impairment from the common phase error [20-21] dominated phase noise after LE....

References

- [20] Wu, S. & Bar-Ness, Y. OFDM systems in the presence of phase noise: consequences and solutions. *IEEE Transactions on Communications* **52**, 1988-1996 (2004).
- [21] Armada, A. G. Understanding the effects of phase noise in orthogonal frequency division multiplexing (OFDM). *IEEE transactions on broadcasting*. **47**, 153-159 (2001).

Reviewer #2 (Remarks to the Author):

The paper demonstrates the use of PICs towards generation of dual optical lines for THz generation. Then, the transmission of 0.4 THz channels is evaluated. The paper combines different approaches for Tx (PIC chip) and post-processing steps (for Rx) somehow already discussed in the literature, by authors. This contribution, while being a combination of many devices/systems sounds more incremental to me than completely new approach.

Authors' response: We would like to thank the Reviewer for the comment.

We agree with the Reviewer that we combine many state-of-the-art devices/subsystems/processing techniques, as we aim at pushing the boundaries with this system-level effort, with the purpose to explore and promote the potential of the adopted hybrid electro-optical THz transmission methodology. In the authors' opinion, the novelty of this contribution is the exploration of the potential integration of advanced technologies in all system segments, as well as the overall system-level performance optimization. Considering all the boundary conditions, it is technically challenging to simultaneously achieve and verify the two key performance metrics we chose to focus on, i.e., 131 Gbit/s single-channel net rate and 10.7-m wireless distance, in a single system with a PIC chip-enabled THz transmitter. To reach our goal, we have performed detailed characterization and overall optimization of the system, combining the injection-locked dual-DFB PIC for high-quality THz carrier generation, 16-QAM-OFDM modulation format, coherent heterodyne detection, and nonlinear equalization-based DSP. We consider such an effort and the achievements result in a significant contribution to the state of the art.

In addition, I feel that several details are missing, for example about the THz devices, for people to be able for full comparisons with other similar works in the field (see my comments).

I have a couple of questions/comments.

The main advantage of the PIC approach as highlighted by authors is the compacity. However, considering just the PIC, the DFB lasers are not locked and a comb frequency is required, thus compacity is more limited.

Q1. How would operate the system with only the PIC chip (without any comb)?

Authors' response: This is an important question, and we are glad to discuss it.

As the Reviewer pointed out, the goal is to achieve frequency-stable and phase-locked optical tones from the dual-DFB PIC. The adopted comb-based optical injection locking scheme in this work is only one of the possible approaches for optical phase-locking. Other on-chip optical phase-locked methods can also be used for stable beating, such as monolithically integrated optical phase-locked loop (OPLL) PIC [1-3].

Additionally, though challenging and not yet demonstrated, to our knowledge, it is not technically prohibitive to further integrate the frequency comb with the dual-DFB PIC, even monolithically. Several chip-based optical frequency combs have been reported, such as on-chip mode-locked laser (MLL) based on generic InP and III-V-on-silicon photonic integration platforms [4], on-chip mode-locked Kerr combs based on the Si₃N₄ micro-resonator [5], and III-V-on-silicon-nitride MLL chip [6].

Therefore, one can expect a potential fully-integrated PIC with either the OPLL- or the frequency comb-based scheme to support the compacity advantage of the PIC approach. Due to the limitation of the experimental condition, we used an off-the-shelf MLL for comb generation in this experiment for proof-of-concept purposes. In our next phase of research, we aim for a monolithically integrated dual-DFB PIC with effective phase-locking, using one of the possible approaches mentioned above.

- [1] Ristic, S., Bhardwaj, A., Rodwell, M. J., Coldren, L. A. & Johansson, L. A. An optical phase-locked loop photonic integrated circuit. *J. Lightw. Technol.* **28**, 526–538 (2010).
- [2] Tetsumoto, T., Nagatsuma, T., Fermann, M.E. et al. Optically referenced 300 GHz millimetre-wave oscillator. *Nat. Photon.* **15**, 516–522 (2021).
- [3] Hasanuzzaman, G. K. M., Haymen Shams, Cyril C. Renaud, John Mitchell, Alwyn J. Seeds, and Stavros Iezekiel. "Tunable THz signal generation and radio-over-fiber link based on an optoelectronic oscillator-driven optical frequency comb." *Journal of Lightwave Technology* **38**, no. 19 (2020): 5240-5247.
- [4] Gasse, K. V., Uvin, S., Moskalenko, V., Latkowski, S., Roelkens, G., Bente, E. & Kuyken, B. Recent advances in the photonic integration of mode-locked laser diodes. *IEEE Photon. Technol. Lett.* **31**, 1870-1873 (2019).
- [5] Jin, W. et al. Hertz-linewidth semiconductor lasers using CMOS-ready ultra-high-Q microresonators. *Nature Photon.* **15**, 346-353 (2021).
- [6] Cuyvers, S. et al. Low noise heterogeneous III-V-on-silicon-nitride mode-locked comb laser. *Laser Photonics Rev.* **15**, 2000485-1-9 (2021).

Revision: Page 13, 2nd paragraph, the corresponding modification in the revised version is as follows:

... It is worth noting that the adopted comb-based optical injection locking scheme in this work is only one of the possible approaches for optical phase-locking to achieve frequency stable and phase-locked optical tones from the dual-DFB PIC. Other on-chip optical phase-locked methods can also be used for a stable beating, such as monolithically integrated optical phase-locked loop (OPLL) PIC [30]. Additionally, though challenging and not yet demonstrated, it is not technically prohibitive to further integrate the frequency comb with the dual-DFB PIC, even monolithically. Several chip-based optical frequency combs have been reported, such as on-chip mode-locked laser (MLL) based on generic InP and III-V-on-silicon photonic integration platforms [31], on-chip mode-locked Kerr combs based on the Si₃N₄ microresonator [32], and III-V-on-silicon-nitride MLL chip [33]. Therefore, one can expect a potential fully integrated PIC with either the OPLL- or the frequency comb-based scheme to support the compacity advantage of the PIC approach. Due to the limitation of the experimental condition, we used an off-the-shelf MLL for comb generation in this experiment for proof-of-concept purposes. In our future research, we aim for a monolithically integrated dual-DFB PIC with effective phase-locking.

References:

- [30] Ristic, S., Bhardwaj, A., Rodwell, M. J., Coldren, L. A. & Johansson, L. A. An optical phase-locked loop photonic integrated circuit. *J. Lightw. Technol.* **28**, 526–538 (2010).

[31] Gasse, K. V., Uvin, S., Moskalenko, V., Latkowski, S., Roelkens, G., Bente, E. & Kuyken, B. Recent advances in the photonic integration of mode-locked laser diodes. *IEEE Photon. Technol. Lett.* **31**, 1870-1873 (2019).

[32] Jin, W. et al. Hertz-linewidth semiconductor lasers using CMOS-ready ultra-high-Q microresonators. *Nature Photon.* **15**, 346-353 (2021).

[33] Cuyvers, S. et al. Low noise heterogeneous III-V-on-silicon-nitride mode-locked comb laser. *Laser Photonics Rev.* **15**, 2000485-1-9 (2021).

It would have been interesting to describe more on the UTC-PD used, the power level and I would have expected to see a link budget to be able to compare with other approaches.

Q2. System-level figures should be included and deeper description of THz Tx and Rx (which I assume to be commercial ones?)

Authors' response: Thank you for the comments on these two system aspects. We address them respectively as follows.

1) Link budget

The system budget is shown in the following figure. The input optical power into the UTC-PD is +14 dBm, and the conversion efficiency is 0.15 A/W. After heterodyne detection in the UTC-PD, the power of the output THz signal is around -24 dBm. The free space path loss (FSPL) is $10 \cdot \log(4 \cdot \pi \cdot 10.7 \cdot 408 \cdot 10^9 / 3 \cdot 10^8)^2 \approx 105$ dB. The total antenna gain of the two antenna/lens combinations is 103 dBi, which is estimated by comparing the received power with the calculated FSPL. The power of the received THz signal is $-24 + 103 - 105 = -26$ dBm. The conversion loss of the THz mixer is around 17 dB, and the gain of the intermediate frequency (IF) amplifier (SHF 807B) is 23 dB. Thus, the power of the IF signal is $-26 - 17 + 23 = -20$ dBm. Then, the IF signal is captured by the DSO for signal processing.

Considering the SNR calculation, the THz mixer and the IF amplifier dominate the noise contributions to the received signal. The average noise level of the THz mixer (VDI WR2.2MixAMC) is -151 dBm/Hz. The noise power is $7.94 \cdot 10^{-19} \text{ W/Hz} \cdot 44.43 \text{ GHz} = 3.53 \cdot 10^{-8} \text{ W} = -44.5$ dBm. The SNR of the IF signal before the IF amplifier is around $-26 - (-44.5) = 18.5$ dB. The noise figure (NF) of the IF amplifier is 4 dB, and the SNR of the IF signal after the IF amplifier is around $18.5 - 4 = 14.5$ dB.

4) Link Budget Calculation

The input optical power into the UTC-PD is +14 dBm, and the conversion efficiency is 0.15 A/W^{-1} . After heterodyne detection in the UTC-PD, the power of the output THz signal is around -24 dBm. The free space path loss (FSPL) is $10 \cdot \log(4 \cdot \pi \cdot 10.7 \cdot 408 \text{e}9 / 3 \text{e}8)^2 \approx 105 \text{ dB}$. The total antenna gain of the two antenna/lens combinations is 103 dBi, which is estimated by comparing the received power with the calculated FSPL. The power of the received THz signal is $-24 + 103 - 105 = -26 \text{ dBm}$. The conversion loss of the THz mixer is around 17 dB, and the gain of the intermediate frequency (IF) amplifier (SHF 807B) is 23 dB. Thus, the power of the IF signal is $-26 - 17 + 23 = -20 \text{ dBm}$. Then, the IF signal is captured by DSO for signal processing.

Considering the SNR calculation, the THz mixer and the IF amplifier dominate the noise contributions to the received signal. The average noise level of the THz mixer (VDI WR2.2MixAMC) is $-151 \text{ dBm} \cdot \text{Hz}^{-1}$. The noise power is $7.94 \text{e-}19 \text{ W/Hz} \cdot 44.43 \text{ GHz} = 3.53 \text{e-}8 \text{ W} = -44.5 \text{ dBm}$. The SNR of the IF signal before the IF amplifier is around $-26 - (-44.5) = 18.5 \text{ dB}$. The noise figure (NF) of the IF amplifier is 4 dB, and the SNR of the IF signal after the IF amplifier is around $18.5 - 4 = 14.5 \text{ dB}$.

Supplementary Fig. 5 Link budget of the experiment system.

2) Deeper description of THz Tx and Rx

For the THz transmitter in this experimental system, we employed the commercial UTC-PD (IOD-PMAN-12001) manufactured by NTT. It's an antenna-integrated photomixer integrating a hybrid-absorber UTC-PD and a bow-tie antenna. The UTC-PD has an ultrafast response with a relatively high output power in the THz range. The response of the UTC-PD is determined by the electron transport in the whole structure, where the electrons exhibit velocity overshoot ($3-5 \times 10^7 \text{ cm/s}$) in the carrier collection layer. This is an essential difference from the conventional p-i-n PD [1] [2]. In addition, UTC-PDs generate higher output saturation current due to the reduced space charge effect in the depletion layer, which also results from the high electron velocity in the depletion layer [2]. For the UTC-PD used in this experiment, the THz output power is in the -30 to -20 dBm level, and the typical value is -24 dBm. The frequency bandwidth of THz output is 200-1500 GHz. The wavelength range of optical input is 1540-1560 nm, and the optical input power is up to 15 dBm. DC responsivity of the UTC-PD is 0.08-0.25 A/W,

and the typical value is 0.15 A/W. The reverse voltage (V_{bias}) is -2.5 to -1 V. The photocurrent (I_{ph}) is up to 12 mA, and the typical I_{ph} is 7 mA (V_{bias} is -1V).

At the receiver, a commercial even harmonic mixer from Virginia Diodes (VDI WR2.2MixAMC) is used. Per the manufacturer's specifications, it supports input THz range from 325 GHz to 500 GHz with an IF output bandwidth of ~40 GHz. The conversion loss is around 17 dB. The sensitivity of the THz input is lower than -30 dBm.

[1] Ishibashi, T., Muramoto, Y., Yoshimatsu, T. & Ito, H. Unitraveling-carrier photodiodes for terahertz applications. *IEEE J. Sel. Topics Quantum Electron.* **20**, 3804210 (2014).

[2] Renaud, C. C., Natrella, M., Graham, C., Seddon, J., Dijk, F. V. & Seeds, A. J. Antenna integrated THz uni-traveling carrier photodiodes. *IEEE J. Sel. Topics Quantum Electron.* **24**, 8500111 (2018).

Revision: Page 17, 1st paragraph, **Method-Experiment** part, the corresponding modification in the revised version is as follows:

For the THz transmitter in this system, we employed a UTC-PD as the photomixer. UTC-PD has an ultrafast response with a relatively high output power in the THz range compared with other photomixers. The response of the UTC-PD is determined by the electron transport in the whole structure, where the electrons exhibit velocity overshoot ($3\text{-}5 \times 10^7 \text{ cm}\cdot\text{s}^{-1}$) in the carrier collection layer [19] [36]. In addition, UTC-PDs generate higher output saturation current due to the reduced space charge effect in the depletion layer, resulting from the high electron velocity in the depletion layer [36]. For our experiment, the UTC-PD is a commercial device (IOD-PMAN-12001) manufactured by NTT. It's an antenna-integrated photomixer integrating a hybrid-absorber UTC-PD and a bow-tie antenna. It operates in C-band with maximum input optical power of 15 dBm. The typical DC responsivity is about 0.15 AW^{-1} , and it covers THz output frequency range of 200-1500 GHz per the manufacture's specifications. With our experimental configuration, the emitted THz signal power from the UTC-PD is found to be around -24 dBm. For the THz signal reception, a commercial even harmonic mixer from Virginia Diodes (VDI, WR2.2MixAMC) is used. It supports input THz ranging from 325 GHz to 500 GHz with an IF output bandwidth of ~40 GHz. The sensitivity of the THz input is less than -30 dBm, and the conversion loss is around 17 dB.

References:

[19] Ishibashi, T., Muramoto, Y., Yoshimatsu, T. & Ito, H. Unitraveling-carrier photodiodes for terahertz applications. *IEEE J. Sel. Topics Quantum Electron.* **20**, 3804210 (2014).

[36] Renaud, C. C., Natrella, M., Graham, C., Seddon, J., Dijk, F. V. & Seeds, A. J. Antenna integrated THz uni-traveling carrier photodiodes. *IEEE J. Sel. Topics Quantum Electron.* **24**, 8500111 (2018).

Concerning the overall compacity of the THz system, the approach has to be compared with CMOS or InP/GaAs systems, which has a moderate (InP/GaAs) or high (CMOS) integration level demonstrated also 100 Gbit/s.

Q3. How is the impact of the proposed scheme over other works?

Authors' response: We thank the Reviewer for this comment, and we are pleased to share our opinions.

As the Reviewer stated, there have recently been a few system-level demonstrations with all-electronic approaches based on III-V semiconductor (e.g., InP HEMT, InGaAs mHEMT), Si CMOS, and SiGe HBT BiCMOS technologies, supporting up to 100 Gbit/s data rate. The generated carrier frequencies are mostly limited up to 300 GHz in these demonstrations. We agree with the Reviewer that eventually, we will need to compare the photonics-based approaches with these all-electronic approaches in all aspects, including compacity, power, energy efficiency, bandwidth, frequency range, tunability, etc. We may expect that the all-electronic approaches will win in some of these aspects, whereas the photonics-assisted approaches will win in others. However, in our opinion, it is too early to draw conclusions at this moment, considering the technology readiness levels (TRLs) between the two approaches are far apart. The all-electronic technologies in III-V and Si have accumulated long-term development experience and momentum since the microwave applications and recently extended to the millimeter-wave/sub-millimeter-wave region. The photonics-based approaches, on the contrary, are still in the early phase of research, and improvements in all these aspects are to be explored. If we extrapolate the trend as of today, we believe that the all-electronic transceivers with high compacity and high energy efficiency may be developed in large volume to support applications up to 300 GHz, and the photonics-based schemes with broader bandwidths may have good potential to support higher bitrate scalability in the frequency bands above 300 GHz.

If we narrow the focus to the photonics-based approaches, we would also like to extend our discussions by comparing the III-V and the CMOS-compatible technologies. When comparing III-V semiconductor (e.g., InP/GaAs) and CMOS-compatible (e.g., silicon-on-insulator/silicon-nitride) chips, the key quantities are footprint and functionality. III-V can emit light, thus favorable for making active photonic components, including lasers and photodiodes required for optical heterodyne-based THz generation and detection. Unlike III-V, Si-based platforms cannot generate light, but thanks to CMOS processes' technological maturity, they make possible low-propagation-loss and large-scale integration.

III-V remains the technology platform of choice for efficient light source devices in optical THz telecommunication systems, which motivates the integration of III-V photonic circuits with other photonics and electronics through various heterogeneous chip-bonding and co-integration methods. Heterogeneous integration, i.e., integrating multiple materials into one device, has emerged to combine the strengths of various technology platforms, although it could rely on a technically complex and costly multi-chip assembly solution. Since an early developed monolithic III-V fully integrated chip for a millimeter-wave generation was reported [1], different integrated devices, including hybrid-substrate InP-polymer dual tunable laser chip [2] and silicon photonics [3,4] for THz wave generation, have been demonstrated. Please notice that in silicon photonics-based devices, external and separate lasers are needed that inevitably decrease their compactness.

Our proposed photonic integrated circuit was developed in an easily accessible monolithic III-V generic foundry platform [5] and this paves a way for development of fairly compact and functional photonic THz chip-scale system. Considering integration with silicon photonics and high-speed photodiodes [6,7], further improved system compactness and efficiency can be expected in future developed devices. Also, in terms of enhancing signal quality, carrier stabilization techniques and comb-injection locking (as introduced in the manuscript) despite that the comb was implemented off-chip, have been investigated [8,9]. Future direction is therefore towards an ultimately compact THz chip-scale system which contains

not only a THz signal source but also a signal stabilization scheme, and for such a relatively large-scale integrated circuit that depends on complex, fast electronics to be closely interfaced with optical modulation and detection once again require a higher degree of co-integration with CMOS.

- [1] Carpintero, G. et al. Microwave photonic integrated circuits for millimeter-wave wireless communications. *J. Lightwave Tech.* **32**, 3495-3501 (2014).
- [2] Carpintero, G., Hisatake, S., Felipe, D. et al. Wireless data transmission at terahertz carrier waves generated from a hybrid InP-polymer dual tunable DBR laser photonic integrated circuit. *Sci. Rep.* **8**, 3018 (2018).
- [3] Harter, T., Muehlbrandt, S., Ummethala, S. et al. Silicon-plasmonic integrated circuits for terahertz signal generation and coherent detection. *Nature Photon.* **12**, 625-633 (2018).
- [4] Moon, S. R., Han, S., Yoo, S., Park, H., Lee, W. K., Lee, J. K., Park, J., Yu, K., Cho, S. H. & Kim, J. Demonstration of photonics-aided terahertz wireless transmission system with using silicon photonics circuit. *Opt. Express* **28**, 23397-23408 (2020).
- [5] Smit, M., Leijtens, X., Bente, E., Tol, J. V., Ambrosius, H., Robbins, D., Wale, M., Grote, N. & Schell, M. Generic foundry model for InP-based photonics. *IET Optoelectron.* **5**, 187-194 (2011).
- [6] Hulme, J., Kennedy, M. J., Chao, R., Liang, L., Komljenovic, T., Shi, J., Szafraniec, B., Baney, D. & Bowers, J. E. Fully integrated microwave frequency synthesizer on heterogeneous silicon-III/V. *Opt. Express* **25**, 2422-2431 (2017).
- [7] Jiao, Y., Tol, J. V. D., Pogoretskii, V. et al. Indium phosphide membrane nanophotonic integrated circuits on silicon. *Phys. Status Solidi A* **217**, 1900606 (2020).
- [8] Lo, M., Jia, S., Kong, D., Morioka, T., Oxenløwe, L. K., Hu, H. & Carpintero, G. Foundry-fabricated dual-DFB PIC injection-locked to optical frequency comb for high-purity THz generation. *In Optical Fiber Communication Conference (OFC 2019)*, W1B.3 (OSA, San Diego, 2019).
- [9] Balakier, K., Fice, M. J., Dijk, F. V., Kervella, G., Carpintero, G., Seeds, A. J. & Renaud, C. C. Optical injection locking of monolithically integrated photonic source for generation of high purity signals above 100 GHz. *Opt. Express* **22**, 29404-29412 (2014).

Revision: Page 15, 2nd paragraph in **Discussions**, the corresponding modification in the revised version is as follows:

Finally, it is worth noting that such photonics-based THz technologies are still in their initial research phase. In the long-term, transceiver technologies based on all-electronic approaches, including the III-V semiconductor and silicon-based CMOS technologies, should all be included in performance comparisons with established common metrology standards upon the maturity of the adopted photonics-based THz technologies.

It is claimed that the 131 Gbit/s over 10.7 m is a record in terms of THz power/bitrate/distance. The authors reported 160 Gbit/s over 0.5 m with I think less antenna gains. However, the single-data-rate performance that is often used in THz communications however it is important to compare the energy

consumption of the overall system and the system performance like the real-time bit-error-rate, the latency time, ...

Most of the THz systems make use of signal processing at receiver stage, thus this implies energy consumption. For fig. 1b., I think only the THz transmitter is considered. Thus, if the receiver makes use of complex post-processing steps, it induces some distortion when comparing different systems.

As a consequence, the figure 1.a and b suggests that the proposed scheme is the highest reported performance, but other parameters should be integrated for a global comparison with other approaches/systems. In my opinion, a deeper comparison between systems should be considered, including the receiver and post-processing.

Remark: usually, the figure of merit of any communication system is the energy per bit (pJ/bit) required for a specified performance (BER for example), rather than pJ/bit/m.

As an example, the reference [16] reported a real-time performance, which means no latency time nor post-detection using signal processing. Here, the post-processing is described in the text and contains several steps. This should be considered for fair comparison with other systems as a real-time performance is more appropriate for practical use-cases.

I feel that this is thus not completely fair to compare on the same graph (fig.1b) the work of [16] that has a real-time performance ($BER < 10^{-9}$) and the proposed approach that comes with $BER > 10^{-2}$, because it could lead for the readers to biased conclusions. THz communications is still requiring common metrology standards, that not yet exists, thus the community has to take care of system-comparisons.

Q4. A more comprehensive/deeper comparison should be considered, by including global energy consumption (because at the end we need a full THz system, thus Tx and Rx consumption should be integrated). Somehow the receiver consumption, receiver sensitivity has to be described.

Authors' response: We appreciate the Reviewer's opinions and suggestions. We discuss the two aspects as follow and make revisions accordingly.

On system comparisons: The Reviewer raised a very important concern regarding system performance comparisons. As the Reviewer pointed out, the THz communications, particularly the photonics-assisted schemes, are still in a low technology readiness level, and there are no existing common metrology standards yet established as a benchmark for proposed schemes to compare with. In this respect, we completely agree with the Reviewer that we need to be very careful in claiming 'record' in comparisons. Meanwhile, we also tried to keep a forward-looking perspective when considering practical limitations, such as the complexity and power consumption of the DSP and FEC modules. Currently, effective digital signal processing and channel coding are widely deployed in commercial systems, for both wireless and optical communications. If we take digital coherent optical transceivers containing complex DSP-ASIC and FEC modules as examples, during the past 10 years the energy consumption has dropped from 90 W (5×7" OIF MSA 40G in 2011) to <20 W (400ZR pluggable in 2021) along with 10 times increase in data rate. Therefore, as we are studying the next-generation THz communication technologies that may be demanded at least a decade from today, we have decided not to exclude the possibilities of using the advanced DSP and coding techniques. In our opinion, the complexity and power consumption of these modules as of today should not be considered as constraints. Therefore, since we are not able to speculate the possible practical limitations beyond the current generation, we decided to solely focus on two

fundamental system metrics, i.e., bit rate and distance in our works (including our previous works), and accordingly utilize all possible DSP and coding tools to explore the system limits.

To address the Reviewer's concern, we agree that our previously used terms such as 'record', though bounded with conditions, can be misleading. In our revised manuscript, we have avoided using such terms and rephrased the texts to clarify the conditions under which we have achieved our results.

Moreover, our original purpose of Fig. 1b is to show the latest achievements in the two system metrics (bit rate and distance) we focus on. In this regard, we have tailored the commonly used figure of merit with normalizations in both emitted THz power and distance to better represent these specific subsystem-level performance in the reported demonstrations. Also, since the main novelty we attempted to make in this work was on the transmitter side, whereas the receiver was still based on previous approach, we presented the relevant system values on the transmitter. However, we agree with the Reviewer that such a straightforward presentation can be unfair and misleading. Therefore, to further avoid confusions in terms of system comparisons, we have converted Fig.1b into a table and included elaborated measurement conditions. We also added comments in the text that certain drawbacks in terms of latency are introduced along with the applied DSP and coding schemes, compared with the real-time DSP-free demonstrations that have been reported.

Finally, we would like to note that the current effort focuses on reducing the energy consumption on the transmitter side, and further efforts in reducing the energy consumption of the receiver are essential. It requires dedicated efforts in both hardware and DSP and will be explored in our future works. We have clarified this point in our revised manuscript.

On receiver consumption and sensitivity: Though the signal processing and FEC decoder at the receiver occupies a significant part of the total power consumption and latency, they can effectively mitigate signal impairments induced by both the hardware and the free-space channel, and enable capacity-approaching communications [1-3]. Therefore, as we stated earlier, we explore the fundamental system limits by utilize advanced DSP and coding tools without limiting ourselves with practical constraints. In what follows we elaborate our receiver DSP and FEC decoder configurations.

The baseband signal processing in our scheme includes linear equalization (LE), phase noise compensation (PNC) and nonlinear equalization (NLE). First, the signal after the FFT module passes through the pilot based one-tap LE, which is used to compensate the system linear response and to reduce the system additive noise influence. After LE, the signal is equalized with a least-squares method-based PNC to reduce the impairment from phase noise. Since the carrier linewidth is low, the signal is equalized with a least-squares method-based PNC to reduce the impairment from the common phase error dominated phase noise after LE. After the PNC, the Volterra series nonlinear model based NLE is used for estimating the nonlinear impairment. In our system, the nonlinear distortions mainly come from the saturation effects of optical IQ-MZM, saturation effects of broadband electrical IF amplifier and the high-order harmonics from the mixer. All these nonlinear impairments will introduce inter-carrier interference to the OFDM signal transmission and deteriorate the EVM performance. Thanks to the injection locking scheme in our proposed system, the laser chirp effect is greatly suppressed and the nonlinear impairments from the lasers are reduced.

The channel coding and decoding dominates the baseband computations, it is hitting the implementation wall because it is most computationally demanding baseband process [4, 5]. The power consumption and latency induced by LE, PNC and NLE are negligible comparing with channel coding and decoding. The three central candidate coding schemes for the future B5G and 6G are Turbo, LDPC and polar codes. Although Turbo and LDPC decoders are both executed on data-flow graphs, Turbo decoding is inherently

serial, and LDPC decoding is inherently parallel. In contrast, polar decoding is typically performed on a tree structure and is inherently serial. Due to their parallel nature, LDPC decoders provide higher throughput [6]. In our experiment, the BER threshold is set according to LDPC based soft-decision FEC limit $2.7E-2$. However, the selected FEC limit use 20% coding overhead, it is also most part of the power consumption and latency.

From a general data-link perspective, investigating into the baseband signal processing algorithms could contribute to the increasing knowledge in the research field, although they are comparatively complicated currently [1]. Due to Moore's Law's diminishing effect, limited advances in chip power density and baseband computations are expected from silicon scaling. Thus, incorporating application-specific integrated circuit (ASIC) in a holistic framework is vital for decreasing the time-to-market of THz communication system [5]. There is a joint force between algorithms (especially coding scheme) and ASIC architecture optimization across all blocks to optimize the throughput, power consumption and latency.

- [1] Zhang, L., Pang, X., Jia, S., Wang, S., and Yu, X. Beyond 100 Gb/s Optoelectronic Terahertz Communications: Key Technologies and Directions. *IEEE Communications Magazine* **58**, 34-40 (2020).
- [2] Rodríguez-Vázquez, P., Leinonen, M. E., Grzyb, J., Tervo, N., Parssinen, A., and Pfeiffer, U. R. Signal-processing challenges in leveraging 100 Gb/s wireless THz. In *2020 2nd 6G Wireless Summit (6G SUMMIT)*. (Levi, Finland, 2020).
- [3] Weithoffer, S., Herrmann, M., Kestel, C., and Wehn, N. Advanced wireless digital baseband signal processing beyond 100 Gbit/s. In *2017 IEEE International Workshop on Signal Processing Systems (SiPS)* (Lorient, France, 2017).
- [4] Nagarajan, R., Lyubomirsky, I., and Agazzi, O. Low Power DSP-Based Transceivers for Data Center Optical Fiber Communications (Invited Tutorial). *Journal of Lightwave Technology* **39**, 5221-5231 (2021).
- [5] Kestel, C., and Herrmann, M. When channel coding hits the implementation wall. In *2018 IEEE 10th International Symposium on Turbo Codes & Iterative Information Processing (ISTC)*. (Hong Kong, China, 2018).
- [6] Mansour, M. M., and Shanbhag, N. R. High-throughput LDPC decoders. *IEEE Transactions on Very Large Scale Integration (VLSI) Systems* **11**, 976-996 (2003).

Revision: Page 4, last paragraph of **Introduction**, the corresponding modification in the revised version is as follows:

Fig. 1(b) shows selected THz wireless transmission demonstrations at above 300 GHz (including long-distance (>10 m) demonstrations), revealing the relation of data rate, distance, and transmitter THz energy per bit per distance [9-10, 14-15, 17-18]. One should note that these numbers are only indicative as the measurement conditions in these demonstrations are different in terms of referenced bit error rate (BER) level and the complexity of employed digital signal processing (DSP). More comprehensive comparisons between these demonstrations will need to be performed with common metrology standards, which are yet to be established.

Page 14, 2nd paragraph of **Discussion**, the corresponding modification in the revised version is as follows:

As effective DSP and channel coding are already widely deployed in most commercial systems for both wireless and optical communications, we use an advanced DSP routine and benchmark our transmission performance with soft-decision LDPC-CC FEC threshold in this demonstration. However, one should note that the drawbacks of employing such sophisticated DSP and FEC decoder are more significant receiver power consumption and higher system latency, compared with the systems operating in 'real-time' with virtually error-free performance. Especially, the highly demanding channel coding and decoding dominate the baseband computations. In the experiment, the LDPC-CC FEC limit uses 20% coding overhead and contributes most of the power consumption and latency in practice. Therefore, for specific high-end applications where low latency is critical, one should consider using low-complexity signal processing and light FEC variants at the cost of lowering the system data rate. It is also worth noticing that considering the baseband signal processing, incorporating application-specific integrated circuits (ASIC) in a holistic framework is vital for decreasing the time-to-market of the THz communication system. A joint optimization between algorithms (especially coding scheme) and ASIC architecture across all blocks is expected to optimize the throughput, power consumption, and latency.

Q5. Authors' previous work which seems to use the same type of UTC-PD should be included as well in this comparison (fig 1.b for example) to evaluate the performance increase of the proposed work.

Remark: speaking about a « record » has to come with full analysis with other systems in my opinion.

Authors' response: We thank the Reviewer for the comments and appreciate the Reviewer's opinions. We completely agree with the Reviewer that we need to be very careful in claiming 'record' in comparisons of system performance. In addition, we have added our previous works, which used the same type of UTC-PD, in this comparison in Fig. 1b, such as 150 Gbit/s 8-channel link [R1], 260 Gbit/s 6-channel transmission [R2] and 106 Gbit/s single-channel system [R3].

(b) Comparison of normalized transmitter THz energy of the reported demonstrations.

Frequency band	Emitted THz power (dBm)	Distance (meter)	Bit rate (Gbit/s)	Measurement conditions	Normalized power (J/bit/m)
400 GHz [14]	10	35	1	Real-time @BER 1×10^{-9}	2.9×10^{-13}
300 GHz [18]	0	110	115	Offline @BER 1.25×10^{-2}	9×10^{-17}
300 GHz [17]	-13	100	50	Real-time @BER 9.5×10^{-4}	1.0×10^{-17}
300-500 GHz [R1]	-24	0.5	150	Offline @BER 3.8×10^{-3}	5.3×10^{-17}
300-500 GHz [R2]	-24	0.5	260	Offline @BER 2×10^{-2}	3.1×10^{-17}

400 GHz [R3]	-24	0.5	106	Offline @BER 2×10^{-2}	7.5×10^{-17}
400 GHz [This work]	-24	10.7	131	Offline @BER 2.7×10^{-2}	2.9×10^{-18}

[R1] Yu, X., Jia, S., Hu, H., Galili, M., Morioka, T., Jepsen, P. U. & Oxenløwe, L. K. 160 Gbit/s photonics wireless transmission in the 300-500 GHz band. *APL Photon.* **1**, 081301 (2016).

[R2] Pang, X., Jia, S., Ozolins, O., Yu, X., Hu, H., Marcon, L., Guan, P., Ros, F. D., Popov, S., Jacobsen, G., Galili, M., Morioka, T., Zibar, D. & Oxenløwe, L. K. 260 Gbit/s photonic-wireless link in the THz band. *In IEEE Photon. Conf. (IPC 2016)*, PD1-2 (IEEE, Hawaii, 2016).

[R3] Jia, S., Pang, X., Ozolins, O., Yu, X., Hu, H., Yu, J., Guan, P., Ros, F. D., Popov, S., Jacobsen, G., Galili, M., Morioka, T., Zibar, D. & Oxenløwe, L. K. 0.4 THz photonic-wireless link with 106 Gb/s single channel bitrate. *J. Lightw. Technol.* **36**, 610-616 (2018).

Revision: Page 4, last paragraph and Fig 1(b), the corresponding modification in the revised version is as follows:

... Fig. 1(b) shows selected THz wireless transmission demonstrations at above 300 GHz (including long-distance (>10 m) demonstrations), revealing the relation of data rate, distance, and transmitter THz energy per bit per distance [9-10, 14-15, 17-18].

(b) Frequency band	Emitted THz power (dBm)	Distance (meter)	Bit rate (Gbits ⁻¹)	Measurement conditions	Normalized power (J·bit ⁻¹ ·m ⁻¹)
400 GHz [14]	10	35	1	Real-time @BER 1×10^{-9}	2.9×10^{-13}
300 GHz [18]	0	110	115	Offline @BER 1.25×10^{-2}	9×10^{-17}
300 GHz [17]	-13	100	50	Real-time @BER 9.5×10^{-4}	1.0×10^{-17}
300-500 GHz [9]	-24	0.5	150	Offline @BER 3.8×10^{-3}	5.3×10^{-17}
300-500 GHz [10]	-24	0.5	260	Offline @BER 2×10^{-2}	3.1×10^{-17}
400 GHz [15]	-24	0.5	106	Offline @BER 2×10^{-2}	7.5×10^{-17}
400 GHz [This work]	-24	10.7	131	Offline @BER 2.7×10^{-2}	2.9×10^{-18}

(b) Comparison of selected representative THz transmission demonstrations above 300 GHz in terms of normalized transmitter THz energy.

In the fig. 5, from (d) to (e), the NLE seems to enhance the constellation but the nonlinear impairments of fig.5. (d) already seems limited. I wonder in which extent the NLE is efficient, of this is due to the used color scale for (b),(c),(d),(e).

Q6. The authors should detail the color-scale, is this linear or not? And qualify the improvement in terms of EVM (%) of the constellations?

Authors' response: Thanks for the comments. The color-scale of the constellation graphs is logarithm-scale. The EVM values in (%) units for case 'w/o Eq.', 'LE', 'LE+PNC', 'LE+PNC+NLE' are 37.3%, 24.2%, 20.4%, 17.1%, respectively, they have been added in figure 5 in the revised manuscript. The nonlinear impairments are common for broadband communication system, since the broadband optical and electrical devices always induce nonlinear distortions, which are not always concerned in microwave and millimeter-wave communication systems. In our system, the nonlinear distortions mainly come from the saturation effects of optical IQ-MZM [1], saturation effects of broadband electrical IF amplifier [2] and the high-order harmonics from the mixer [3]. All these nonlinear impairments will introduce inter-carrier interference to the OFDM signal transmission and deteriorate the EVM performance. Thanks to the injection locking scheme in our proposed system, the laser chirp effect is greatly suppressed and the nonlinear impairments from the lasers are reduced.

[1] Yang, H., Zeng, J., Zheng, Y., Jung, H. D., Huiszoon, B., Van Zantvoort, J. H. C., Tangdionga, E., and Koonen, A. M. J. Evaluation of effects of MZM nonlinearity on QAM and OFDM signals in RoF transmitter. *In International Topical Meeting on Microwave Photonics jointly held with the 2008 Asia-Pacific Microwave Photonics Conference*, 90-93. (Gold Coast, QLD, Australia, 2008).

[2] Costa, E., Midrio, M., and Pupolin, S. Impact of amplifier nonlinearities on OFDM transmission system performance. *IEEE Communications Letters* **3**, 37-39 (1999).

[3] Mehdi, I., Siles, J. V., Lee, C., and Schlecht, E. THz diode technology: Status, prospects, and applications. *Proceedings of the IEEE* **105**, 990-1007 (2017).

Revision: Page 11, Figure 5 part

Fig. 5. BER versus the optical power launched into the UTC-PD for 4 cases with different combinations of DSP modules. (b-e) Constellations for 4 cases with 14 dBm optical power and different combinations of DSP modules (color in logarithm-scale). (f) The electrical spectrum of the 16-QAM-OFDM signal before down conversion. (g) The SNR versus the frequency after down-conversion.

The overall BER values obtained are high and higher than 10^{-2} , even with complex post-processing, which means that practical implementation.

Q7. What is the main limiting factor in this system and what would be the perspectives to make this approach a reality for use-case of fig. 2?

Authors' response: Thank the Reviewer for this value comment.

The main limiting factors of the system in this manuscript are the noise level of the THz receiver and the low emitted power of the THz transmitter (UTC-PD). The current system is only a principled exploration and in the future work we will improve the PIC chips and devices for the practical applications.

In the future, to make this approach a reality for use-case of fig. 2, the perspectives would include on-PIC chip optical frequency comb generation, integrated multiple-lasers chip, integrated optical broadband modulator, couplers, amplifiers and filters, integrated THz transmitter with enhanced output power, highly directional THz beamforming and beam steering, integrated low-noise and wide-bandwidth THz amplifiers for both transmitter and receiver with more efficiency, and THz antennas with higher gains [1][2].

In details, PIC chip-based optical frequency combs can be realized, such as on-chip mode-locked lasers (MLLs) based on generic InP/III-V-on-silicon [3], and III-V-on-silicon-nitride [4], and on-chip mode-locked Kerr combs based on the Si_3N_4 micro-resonator [5]. Integrated multiple-lasers chip has been achieved on the III-V PIC platform [6] such as the chip used in this manuscript. Other optical components such as modulator, couplers, amplifiers and filters also have integration solutions [7]. The photonics-based THz phased array supporting more output power, THz beamforming and beam steering, has been developed towards monolithic integration [8].

[1] Nagatsuma, T., Ducournau, G. & Renaud, C. C. Advances in terahertz communications accelerated by photonics. *Nature Photon.* **10**, 371-379 (2016).

[2] Sengupta, K., Nagatsuma, T. & Mittleman, D. M. Terahertz integrated electronic and hybrid electronic–photonic systems. *Nature Electron.* **1**, 622-635 (2018).

[3] Gasse, K. V., Uvin, S., Moskalenko, V., Latkowski, S., Roelkens, G., Bente, E. & Kuyken, B. Recent advances in the photonic integration of mode-locked laser diodes. *IEEE Photon. Technol. Lett.* **31**, 1870-1873 (2019).

[4] Cuyvers, S. et al. Low noise heterogeneous III-V-on-silicon-nitride mode-locked comb laser. *Laser Photonics Rev.* **15**, 2000485-1-9 (2021).

[5] Jin, W. et al. Hertz-linewidth semiconductor lasers using CMOS-ready ultra-high-Q microresonators. *Nature Photon.* **15**, 346-353 (2021).

[6] Lo, M., Zarzuelo, A., Guzman, R. & Carpintero, G. Monolithically integrated microwave frequency synthesizer on InP generic foundry platform. *J. Lightwave Technol.* **36**, 4626-4632 (2018).

[7] Hu, H. et al. Single-source chip-based frequency comb enabling extreme parallel data transmission. *Nature Photon.* **12**, 469-473 (2018).

[8] Che, M., Matsuo, Y., Kanaya, H., Ito, H., Ishibashi, T. & Kato, K. Optoelectronic THz-wave beam steering by arrayed photomixers with integrated antennas. *IEEE Photon. Technol. Lett.* **32**, 979-982 (2020).

Revision: Page 14, 1st paragraph, the corresponding modification in the revised version is as follows:

The main limitations of the communication system are the noise level of the THz receiver and the low emitted power of the UTC-PD. In the future, to make this approach a reality for practical applications, the perspectives would include on-PIC chip optical frequency comb generation, integrated multiple lasers chip, integrated optical modulator, couplers and filters, integrated THz transmitter with more output power, highly directional THz beamforming and beam steering [2][34]. In detail, PIC chip-based optical frequency combs can be realized, such as on-chip MLLs [31-33]. Integrated multiple lasers chip has been achieved on the III-V PIC platform [6]. Other optical components also have integration solutions [28]. The photonics-based THz phased array supporting more output power, THz beamforming, and beam steering has been developed towards monolithic integration [35].

References:

- [34] Sengupta, K., Nagatsuma, T. & Mittleman, D. M. Terahertz integrated electronic and hybrid electronic–photonic systems. *Nature Electron.* **1**, 622-635 (2018).
- [35] Che, M., Matsuo, Y., Kanaya, H., Ito, H., Ishibashi, T. & Kato, K. Optoelectronic THz-wave beam steering by arrayed photomixers with integrated antennas. *IEEE Photon. Technol. Lett.* **32**, 979-982 (2020).

Reviewer #3 (Remarks to the Author):

This manuscript describes a photonic integrated circuit coupled to an optical frequency comb, which produces two high-purity tones from two injection-seeded DFB lasers. These are then mixed to produce a narrowband tone near 400 GHz. One of the optical tones can be modulated first, thus producing a high-data-rate modulation on the THz signal. The work represents a significant step in the demonstration of photonics-based wireless links in the THz regime. The achieved results are impressive, and will attract a great deal of attention from the community. Therefore I recommend publication, after the authors consider a couple of small (but important) modifications.

1. The abstract suggests that the result at 10m range was achieved with a rather low THz output power (-24 dBm?). This is a bit surprising, so it would be valuable if the authors included an analysis of the link budget (this could possibly be put in the supplementary materials, if there is no room in the main text). What's the free-space path loss at this frequency? What are your antenna gains? In light of that link budget analysis, does it make sense that you are able to achieve ~13 dB of SNR? Well, but that SNR is after all of the DSP, so then what should the link budget be compared to? These considerations are important for understanding how to extrapolate this system demonstration to, e.g., longer range.

Authors' response: Thanks for the comments. The link budget is calculated as follows. The system budget is shown in the following figure. The input optical power into the UTC-PD is +14 dBm, and the conversion efficiency is 0.15 A/W. After heterodyne detection in the UTC-PD, the power of the output THz signal is around -24 dBm. The free space path loss (FSPL) is $10 \cdot \log(4 \cdot \pi \cdot 10.7 \cdot 408e9 / 3e8)^2 \approx 105$ dB. The total antenna gain of the two antenna/lens combinations is 103 dBi, which is estimated by comparing the received power with the calculated FSPL. The power of the received THz signal is $-24 + 103 - 105 = -26$ dBm. The conversion loss of the THz mixer is around 17 dB and the gain of the intermediate frequency (IF) amplifier (SHF 807B) is 23 dB. Thus, the power of the IF signal is $-26 - 17 + 23 = -20$ dBm. Then, the IF signal is captured by DSO for signal processing.

Considering the SNR calculation, the THz mixer and IF amplifier contributes most of the system noise. The average noise level of the THz mixer (VDI WR2.2MixAMC) is -151 dBm/Hz. The noise power is $7.94e-19$ W/Hz $\cdot 44.43$ GHz = $3.53e-8$ W = -44.5 dBm. The SNR of the IF signal before the IF amplifier is around $-26 - (-44.5) = 18.5$ dB. The noise figure (NF) of the IF amplifier is 4 dB, and the SNR of the IF signal after the IF amplifier is around $18.5 - 4 = 14.5$ dB. After receiver side DSP, the SNR is around 13.45 dB.

Revision: Page 30, Supplementary Information Part

4) Link Budget Calculation

The input optical power into the UTC-PD is +14 dBm, and the conversion efficiency is 0.15 A/W^{-1} . After heterodyne detection in the UTC-PD, the power of the output THz signal is around -24 dBm. The free space path loss (FSPL) is $10 \cdot \log(4 \cdot \pi \cdot 10.7 \cdot 408 \text{e}9 / 3 \text{e}8)^2 \approx 105 \text{ dB}$. The total antenna gain of the two antenna/lens combinations is 103 dBi, which is estimated by comparing the received power with the calculated FSPL. The power of the received THz signal is $-24 + 103 - 105 = -26 \text{ dBm}$. The conversion loss of the THz mixer is around 17 dB, and the gain of the intermediate frequency (IF) amplifier (SHF 807B) is 23 dB. Thus, the power of the IF signal is $-26 - 17 + 23 = -20 \text{ dBm}$. Then, the IF signal is captured by DSO for signal processing.

Considering the SNR calculation, the THz mixer and the IF amplifier contributes most of the system noise. The average noise level of the THz mixer (VDI WR2.2MixAMC) is $-151 \text{ dBm} \cdot \text{Hz}^{-1}$. The noise power is $7.94 \text{e-}19 \text{ W/Hz} \cdot 44.43 \text{ GHz} = 3.53 \text{e-}8 \text{ W} = -44.5 \text{ dBm}$. The SNR of the IF signal before the IF amplifier is around $-26 - (-44.55) = 18.5 \text{ dB}$. The noise figure (NF) of the IF amplifier is 4 dB, and the SNR of the IF signal after the IF amplifier is around $18.5 - 4 = 14.5 \text{ dB}$.

Supplementary Fig. 5 Link budget of the experiment system.

2. It is clear from Fig. 5 that the nonlinear equalization and phase noise compensation steps are critical to obtaining a good BER result. What is less clear is why that's so. There is a nice detailed discussion of the signal processing approach in the supplementary materials; but, it would be very nice if the authors could include in the main text some brief discussion of why these steps make such a big difference. Given the Hz-level linewidth and the SSB noise data presented in Fig. 3, I would not have expected a great deal of phase noise, and therefore would not have expected phase noise compensation to have made such a big difference in the constellation diagram. So, can one easily understand why this is so critical? Also, the nonlinearity that gives rise to the requirement for nonlinear equalization presumably originates mostly from the UTC-PD, correct? Or is there some other important source of nonlinearity here? In the interests of improving the accessibility of the manuscript to a broad audience, these issues deserve some discussion.

Authors' response: Thanks for the comments. The influence from the phase noise and nonlinearity are discussed here.

1) Phase Noise Effect

The influence of the linewidth and the BER mainly depends on the phase noise's effect on the OFDM transmission performance. Phase noise $\phi(n)$ in the system can be described as a continuous Brownian motion process with zero mean and variance σ_ϕ^2 , which, from a spectral point of view, can be presented as a finite-power Wiener process [1]. The value of the variance σ_ϕ^2 can be calculated as the integration of the phase noise spectrum $S(f)$ [2], which is:

$$\sigma_\phi^2 = \int S(f)df .$$

Here, we use the phase noise spectrum under 12 dBm comb-injection power, which is shown in the following figure.

Then, the phase noise variance σ_ϕ^2 is calculated to be 0.09503 rad^2 . According to [3], the signal-to-noise ratio (SNR) degradation degrSNR follows the expression:

$$\text{degrSNR} = 10 \cdot \log(1 + \sigma_\phi^2 \cdot \text{SNR}) \text{ dB} .$$

Therefore, the SNR degradation for $\sigma_\phi^2 = 0.09503 \text{ rad}^2$ is shown in the following figure:

It can be seen that OFDM is quite sensitive to the phase noise, since the phase noise causes leakage of FFT and destroys the orthogonality among subcarrier signals. What's more, the influence can be categorized as common phase error (CPE) and inter-carrier interference (ICI) [4]. Assuming the OFDM symbol is:

$$x(n) = \sum_{k=0}^{N-1} s_k \cdot e^{j(2\pi/N)kn}, \quad n = 0, 1, \dots, N-1,$$

where s_k is the QAM signal, N is the number of subcarriers. Since the phase noise $\phi(n)$ is small, the phase noise effect can be approximately expressed as:

$$e^{j\phi(n)} \approx 1 + j\phi(n).$$

In this case, the demultiplexed signal at the receiver side is:

$$y(k) \approx s_k + \frac{j}{N} \sum_{r=0}^{N-1} s_r \sum_{n=0}^{N-1} \phi(n) \cdot e^{j(2\pi/N)(r-k)n} \approx s_k + c_k.$$

The term c_k is the effect of phase noise on the OFDM signal. When $r=k$, c_k is the CPE interference, we have a common error added to every subcarrier by a rotation of the constellation. Thanks to the common error nature of CPE, it can be easily compensated with least-square PNC method. Comparatively, when $r \neq k$, c_k is the ICI interference, which is the summation of the information of the other $N-1$ subcarriers each multiplied by a set of complex numbers. Because of the random and nonlinear nature of ICI, its compensation is complicated and performance is still getting influenced.

In our scheme, the phase noise bandwidth is limited since the linewidth of the carrier is narrow, so the CPE dominates over ICI in our scheme. Thus, a simple PNC scheme could compensate the phase noise effect of the OFDM frames and reduce the performance degradation, which can be observed in the constellation graphs in the paper (Fig. 5). What's more, the NLE could further compensate the ICI induced nonlinear impairments and improve the performance.

In the revised manuscript, we have added more reference papers to give the readers more information about the effect of linewidth on the system performance.

2) Nonlinear Effect

The nonlinear impairments are common for broadband communication system, since the broadband optical and electrical devices always induce nonlinear distortions, which are not always concerned in

microwave and millimeter-wave communication systems. In our system, the nonlinear distortions mainly come from the saturation effects of optical IQ-MZM [5], saturation effects of broadband electrical IF amplifier [6] and the high-order harmonics from the mixer [7]. All these nonlinear impairments will introduce inter-carrier interference to the OFDM signal transmission and deteriorate the EVM performance. Thanks to the injection locking scheme in our proposed system, the laser chirp effect is greatly suppressed and the nonlinear impairments from the lasers are reduced.

The square-law detection effect of the photodiode will introduce nonlinear impairments in the baseband signal, like subcarrier-to-subcarrier beating interference (SSBI) in OFDM signal transmission [8]. However, the UTC-PD is used for heterodyne detection between the optical LO carrier at wavelength λ_1 and optical baseband signal at wavelength λ_2 , the baseband SSBI is out of the response range of the UTC-PD and its connected antenna. Therefore, the UTC-PD is not the main source of the nonlinearity of our system.

- [1] Wu, S. & Bar-Ness, Y. OFDM systems in the presence of phase noise: consequences and solutions. *IEEE Transactions on Communications* **52**, 1988-1996 (2004).
- [2] Robins, W. P. Phase noise in signal sources: theory and applications. **9**, IET (1984).
- [3] Moeneclaey, M. The effect of synchronization errors on the performance of orthogonal frequency-division multiplexed (OFDM) systems. *In Proc. COST 254'Emerging techniques for Communication Terminals'*. (Toulouse, France, 1997).
- [4] Armada, A. G. Understanding the effects of phase noise in orthogonal frequency division multiplexing (OFDM). *IEEE transactions on broadcasting* **47**, 153-159 (2001).
- [5] Yang, H., Zeng, J., Zheng, Y., Jung, H. D., Huiszoon, B., Van Zantvoort, J. H. C., Tangdionga, E., and Koonen, A. M. J. Evaluation of effects of MZM nonlinearity on QAM and OFDM signals in RoF transmitter. *In International Topical Meeting on Microwave Photonics jointly held with the 2008 Asia-Pacific Microwave Photonics Conference*, 90-93 (Gold Coast, QLD, Australia, 2008).
- [6] Costa, E., Midrio, M., and Pupolin, S. Impact of amplifier nonlinearities on OFDM transmission system performance. *IEEE Communications Letters* **3**, 37-39 (1999).
- [7] Mehdi, I., Siles, J. V., Lee, C., and Schlecht, E. THz diode technology: Status, prospects, and applications. *Proceedings of the IEEE* **105**, 990-1007 (2017).
- [8] Peng, W. R., Morita, I., and Tanaka, H. Enabling high capacity direct-detection optical OFDM transmissions using beat interference cancellation receiver. *In 36th European Conference and Exhibition on Optical Communication*. (Turin, Italy, 2010).

Revision: Page 11, 2nd paragraph, the corresponding modification in the revised version is as follows:

... Since the carrier linewidth is low, the signal is equalized with a least-squares method-based PNC to reduce the impairment from the common phase error [20-21] dominated phase noise after LE.

References:

[20] Wu, S. & Bar-Ness, Y. OFDM systems in the presence of phase noise: consequences and solutions. *IEEE Transactions on Communications* **52**, 1988-1996 (2004).

[21] Armada, A. G. Understanding the effects of phase noise in orthogonal frequency division multiplexing (OFDM). *IEEE transactions on broadcasting*. **47**, 153-159 (2001).

Page 11, 2nd paragraph, the corresponding modification in the revised version is as follows:

... In our system, the nonlinear distortions mainly come from the saturation effects of optical IQ-MZM [22], saturation effects of broadband electrical IF amplifier [23], and the high-order harmonics from the mixer [24]. All these nonlinear impairments will introduce inter-carrier interference to the OFDM signal transmission and deteriorate the EVM performance. Thanks to the injection locking scheme in our proposed system, the laser chirp effect is significantly suppressed, and the nonlinear impairments from the lasers are reduced.

References:

[22] Yang, H., Zeng, J., Zheng, Y., Jung, H. D., Huiszoon, B., Van Zantvoort, J. H. C., Tangdionga, E., and Koonen, A. M. J. Evaluation of effects of MZM nonlinearity on QAM and OFDM signals in RoF transmitter. *In International Topical Meeting on Microwave Photonics jointly held with the 2008 Asia-Pacific Microwave Photonics Conference*, 90-93. (Gold Coast, QLD, Australia, 2008).

[23] Costa, E., Midrio, M., and Pupolin, S. Impact of amplifier nonlinearities on OFDM transmission system performance. *IEEE Communications Letters* **3**, 37-39 (1999).

[24] Mehdi, I., Siles, J. V., Lee, C., and Schlecht, E. THz diode technology: Status, prospects, and applications. *Proceedings of the IEEE* **105**, 990-1007 (2017).

3. Just out of curiosity: why 408GHz? Looks like you could have chosen just about any frequency you wanted, given the span of your OFC. Wouldn't you have gotten better performance at, e.g., 340 GHz, with lower FSPL?

Authors' response: We thank the Reviewer for this question.

In the long-term use of this THz transmitter and receiver (320-500 GHz) for doing experiments, we have found that the performance of the THz receiver in the frequency band around 400 GHz is the best. Therefore, we can find the optimal result of the transmission performance in the 400-GHz frequency band. The use of this frequency band is our optimized result after the experimental optimization process. Due to the limitation of THz receiver's bandwidth (320-500 GHz), the transmission system has to be operated within this frequency range. In the 320-500 GHz frequency band, the free-space path loss (FSPL) at each carrier frequency can be almost compensated by THz lenses within the Rayleigh length of the lenses. Meanwhile, the emitting power of the UTC-PD as THz transmitter is not much different within this

frequency range. Hence the overall transmission system performance is determined by the performance of THz receiver. Therefore, after the experimental optimization process, we finally chose 408 GHz as the final wireless carrier frequency.

Revision: Page 10, 3rd paragraph, the corresponding modification in the revised version is as follows:

... It is noted that the wireless carrier frequency of 408 GHz was chosen as the THz receiver is found to perform the best in the frequency band around 400 GHz.

REVIEWERS' COMMENTS

Reviewer #2 (Remarks to the Author):

Authors made a big effort to respond to all reviewer's questions. I am fine now with this version.

Reviewer #3 (Remarks to the Author):

In my opinion, the detailed response document provided by the authors does indeed adequately address all of the concerns expressed in the referee reports.